# Unveiling Impact of Frequency Components on Membership Inference Attacks for Diffusion Models

## Abstract

Diffusion models have achieved tremendous success in image generation, but they also raise significant concerns regarding privacy and copyright issues. Membership Inference Attacks (MIAs) are designed to ascertain whether specific data were utilized during a model's training phase. As current MIAs for diffusion models typically exploit the model's image prediction ability, we formalize them into a unified general paradigm which computes the membership score for membership identification. Under this paradigm, we empirically find that existing attacks overlook the inherent deficiency in how diffusion models process high-frequency information. Consequently, this deficiency leads to member data with more high-frequency content being misclassified as hold-out data, and hold-out data with less high-frequency content tend to be misclassified as member data. Moreover, we theoretically demonstrate that this deficiency reduces the membership advantage of attacks, thereby interfering with the effective discrimination of member data and hold-out data. Based on this insight, we propose a plug-and-play high-frequency filter module to mitigate the adverse effects of the deficiency, which can be seamlessly integrated into any attacks within the general paradigm without additional time costs. Extensive experiments corroborate that this module significantly improves the performance of baseline attacks across different datasets and models.

## 1 Introduction

Diffusion models (Song et al., 2020a; Ho et al., 2020) have achieved significant success in areas such as image generation (Choi et al., 2021; Si et al., 2024; Huberman-Spiegelglas et al., 2024) and video generation (Wu et al., 2024; Blattmann et al., 2023), and have been widely applied in real life. However, this success has brought increasing attention to copyright and privacy issues from both academia and industry (Wang et al., 2023; Wen et al., 2024; Zhang et al., 2024a). Recent research shows that diffusion models exhibit a strong memory effect regarding images in the training set, making the risk of privacy leakage a serious concern (Wen et al., 2024; Duan et al., 2023).

Membership Inference Attacks (MIAs) are crucial for assessing model privacy. Their core objective is to determine whether specific data were utilized during a model's training phase (Shokri et al., 2017). Generally, MIAs exploit the overfitting characteristics of models. They achieve this by capturing the discrepancies in how the model fits the training data compared to other data, thereby enabling the execution of these attacks (Yeom et al., 2018). In the field of image generation, there has been extensive prior research on MIAs targeting Variational Autoencoders (VAEs) (Hilprecht et al., 2019) and Generative Adversarial Networks (GANs) (Chen et al., 2020; Hayes et al., 2017). However, due to the distinct training and generation mechanisms of diffusion models, these established attacks are mostly ineffective when applied to diffusion models (Duan et al., 2023).

Recently, MIAs for diffusion models have received increasing attention. Matsumoto et al. (2023) proposed a query-based attack strategy that determines member data by analyzing the model's loss function. SecMI (Duan et al., 2023) is based on DDIM inversion (Song et al., 2020a; Kim et al., 2022), obtaining intermediate outputs during generation through a deterministic inversion process. Kong et al. (2023) introduced a proximal initialization method that utilizes model predictions to obtain initial noise. Zhai et al. (2024) studied text-to-image diffusion models, exploring the associative

memory between texts and images based on likelihood estimation. In summary, the core mechanism of these methods is to quantify the model's image recovery ability and use this as the basis for constructing the membership inference decision logic.

From the perspective of frequency principles, diffusion models exhibit a distinctive generation process: they first denoise low-frequency signals representing the overall structure and subsequently incorporate high-frequency details into the samples (Yang et al., 2023; Falck et al., 2025). This fundamental processing asymmetry means diffusion models handle low-frequency components with greater fidelity and consistency, while high-frequency components show more variation in their reconstruction. While current membership inference attacks have significantly advanced the field, they have not explicitly considered how this frequency-dependent processing affects their effectiveness. This gap is important because a diffusion model's varying behaviour across frequency bands directly influences its distinctive processing of training versus non-training images, the exact signal MIAs seek to detect. Our analysis within a general error-based MIA paradigm revealed that high-frequency components introduce substantial standard deviation in membership scores, often leading to the misclassification of certain samples. We term this phenomenon "high-frequency deficiency".

Based on the above observations and analyses, we propose a plug-and-play high-frequency filter module. This module exhibits broad applicability and can be integrated into all attacks within the defined general paradigm. Specifically, we transform existing attacks into a metric for quantifying the distance between the target and predicted images. We leverage the Fourier transform to convert the image from the spatial domain to the frequency domain and subsequently apply a filtering operation to selectively remove the high-frequency information. By eliminating the standard deviation caused by high-frequency deficiency, we effectively enhance the performance of existing attacks with negligible additional time overhead. Our contributions can be summarized as follows:

- To the best of our knowledge, this study is the first to explore the impact of frequency domain information on MIAs targeting diffusion models. We formalize a general paradigm for existing error-based attacks and conduct an in-depth analysis of the impact of frequency domain information. The results reveal that existing attacks generally overlook the standard deviation of membership score induced by high-frequency deficiency, which restricts their attack performance.

- To address this issue, we proposed a plug-and-play high-frequency filter module. This module effectively suppresses high-frequency deficiency, and we theoretically demonstrated its capacity to improve attack intensity. This module can be seamlessly integrated into all error-based attacks with negligible additional time overhead.

- We conducted extensive experiments to validate the effectiveness of our method. The results indicate that the high-frequency filter significantly improves the performance of existing attacks, achieving substantial improvements in key metrics such as Attack Success Rate (ASR), Area Under the Curve (AUC), and the True Positive Rate at 1% False Positive Rate (TPR@1% FPR).

## 2 RELATED WORKS

**Membership Inference Attacks.** Shokri et al. (2017) proposed the membership inference attacks, which primarily targeted classification models in machine learning. As the evolution of membership inference attacks continues, they can be classified into two categories: black-box attacks (Salem et al., 2018; Song & Mittal, 2021; Choquette-Choo et al., 2021) and white-box attacks (Nasr et al., 2019; Leino & Fredrikson, 2020), determined by the degree of access granted to the target model. In a white-box setting, the attacker can access the model parameters, whereas in a black-box setting, the attacker only receives the final output of the model. Moreover, noteworthy advancements have been achieved in membership inference attacks targeting generative models. Hayes et al. (2017) demonstrated that membership can be effectively discerned through the logits of the discriminator in GANs. Hilprecht et al. (2019) introduced a Monte Carlo scoring methodology incorporating the reconstruction loss term to facilitate attacks on VAEs.

**Membership Inference Attacks on Diffusion Models.** Recently, membership inference attacks on diffusion models have garnered increasing attention. In white-box settings, Pang et al. (2023) proposed executing an attack through the utilization of gradient information extracted from loss. In grey-box settings, the attacker can only access the intermediate and final outputs (Duan et al., 2023). Matsumoto et al. (2023) pioneered the approach of employing diffusion loss to perform

query-based membership inference. Duan et al. (2023) introduced an attack leveraging DDIM inversion to retrieve intermediate outputs from the models. Meanwhile, Kong et al. (2023) proposed a proximal initialization technique to acquire the deterministic initial noise. The attack is realized by the prediction of this noise from the model. In addition, attacks leveraging the correlation between texts and images have also achieved advancements (Wen et al., 2024; Zhai et al., 2024; Li et al., 2024). Moreover, there has also been a growing interest in attacks for diffusion models in black-box settings (Pang & Wang, 2023).

**Frequency Analysis for Diffusion Models.** Rissanen et al. (2022) observed the implicit spectral bias of diffusion models, which favors generating low-frequency components before high-frequency components. Yang et al. (2023) has also demonstrated, from the perspective of the power spectrum of natural images, that diffusion models initially learn to recover low-frequency components, and then, in fewer subsequent time steps, learn to recover high-frequency components. This results in greater uncertainty in the restoration of high-frequency details. Falck et al. (2025) empirically verifies, from a signal-to-noise ratio perspective, that during training, high-frequency components are more rapidly and early masked compared to low-frequency components. This leads to instability and greater variation in the quality of high-frequency generation. In addition, this inherent low-to-high frequency generation pattern observed in diffusion models is evident in both text-to-image and video generation (Yi et al., 2024; Zhang et al., 2024b).

## 3 PRELIMINARIES

To understand how frequency information affects diffusion models' behaviour in membership inference contexts, we first establish the fundamental mechanisms of diffusion models and how images can be represented in the frequency domain.

**Denoising Diffusion Implicit Model (DDIM).** DDIM (Song et al., 2020a) upgrades the DDPM (Ho et al., 2020) framework by incorporating a non-Markovian process, which effectively decouples $x_{t-1}$ from $x_t$. This innovation allows for skipping timesteps, significantly accelerating the sampling process. DDIM redefines the denoising distribution as follows:

$$p_\theta(x_{t-1}|x_t) = \mathcal{N}\left(\sqrt{\overline{\alpha}_{t-1}}x_0 + \sqrt{1 - \overline{\alpha}_{t-1} - \sigma_t^2} \cdot \frac{x_t - \sqrt{\overline{\alpha}_t}x_0}{\sqrt{1 - \overline{\alpha}_t}}, \sigma_t^2 I\right), \quad (1)$$

where $\overline{\alpha}_t = \prod_{i=1}^t \alpha_i$ and $(\alpha_1, \ldots, \alpha_T)$ are the predefined noise schedules, $\sigma_t$ is the variance schedule. Let $\epsilon_\theta(x_t, t)$ denote the noise predicted by the denoising network. The denoising process is as follows:

$$x_{t-1} = \sqrt{\overline{\alpha}_{t-1}}\left(\frac{x_t - \sqrt{1 - \overline{\alpha}_t}\epsilon_\theta(x_t, t)}{\sqrt{\overline{\alpha}_t}}\right) + \sqrt{1 - \overline{\alpha}_{t-1} - \sigma_t^2}\epsilon_\theta(x_t, t) + \sigma_t\epsilon, \quad (2)$$

where $\epsilon \sim \mathcal{N}(0, I)$, $\sigma_t = \eta\sqrt{(1 - \overline{\alpha}_{t-1})/(1 - \overline{\alpha}_t)}\sqrt{1 - \overline{\alpha}_t/\overline{\alpha}_{t-1}}$, $\eta \in [0, 1]$. The case $\eta = 0$ corresponds to the deterministic DDIM, while $\eta = 1$ corresponds to the DDPM. *More details about the DDPM is provided in Appendix A.1.*

**Frequency Domain Representation of Images.** Frequency domain analysis decomposes an image according to a set of basis functions. We focus on the Fourier transform. For simplicity, we only introduce the formulation for grey images, while it is extendable to multi-channel images. Low-frequency components generally correspond to an image's overall structure and smooth regions, while high-frequency components represent details and edges. Given a $H \times W$ input signal $\mathbf{x} \in \mathbb{R}^{H \times W}$, Discrete Fourier Transform (DFT) projects it onto a collection of sine and cosine waves of different frequencies and phases:

$$\mathbf{X}(u, v) = FFT(\mathbf{x}) = \sum_{x=1}^H \sum_{y=1}^W \mathbf{x}(x, y)e^{-j2\pi\left(\frac{u}{H}x + \frac{v}{W}y\right)}, \quad (3)$$

where $\mathbf{x}(x, y)$ is the pixel value at $(x, y)$; $\mathbf{X}(u, v)$ represents complex value at frequency $(u, v)$; $e$ and $j$ are Euler's number and the imaginary unit. The inverse Fourier transform is denoted as:

$$\mathbf{x}(x, y) = IFFT(\mathbf{x}) = \frac{1}{HW}\sum_{u=1}^H \sum_{v=1}^W \mathbf{X}(u, v)e^{j2\pi\left(\frac{u}{H}x + \frac{v}{W}y\right)}. \quad (4)$$

## 4 METHODOLOGY

### 4.1 FORMALIZATION OF MIAs FOR DIFFUSION MODELS

**Threat Model.** Membership inference focuses on determining whether specific data were utilized in the training. Formally, consider a model $f_\theta$ parameterized by weights $\theta$ and a dataset $D = \{x_1, \ldots, x_n\}$ sampled from data distribution $q_{data}$. Following established conventions (Sablayrolles et al., 2019; Carlini et al., 2022; Duan et al., 2023), $D$ is split into two subsets, $D_M$ and $D_H$. $D_M$ is the member set of $f_\theta$ and $D_H$ is the hold-out set, such that $D = D_M \cup D_H$, $\varnothing = D_M \cap D_H$. So, $f_\theta$ is trained on $D_M$. Each sample $x_i$ is equipped with a membership identifier $m_i$, where $m_i = 1$ if $x_i \sim D_M$; otherwise, $m_i = 0$. The attackers have access to the $f_\theta$ and $D$ but lack knowledge of the specific partitioning between $D_M$ and $D_H$. The goal is to design an attack algorithm $\mathcal{A}$ that predicts the membership identifier $m_i$ for any given sample $x_i$:

$$\mathcal{A}(x_i, \theta) = \mathbb{1} \left[ \mathbb{P}(m_i = 1 \mid \theta, x_i) \geq \tau \right], \tag{5}$$

where $\mathcal{A}(x_i, \theta) = 1$ means $x_i$ comes from $D_M$, $\mathbb{1}[A] = 1$ if $A$ is true, and $\tau$ is the threshold. For generative models, we extend this framework by denoting the generator as $G_\theta$ with weights $\theta$ and the generative distribution as $p_\theta(x)$, where the generated samples $x_i \sim p_\theta(x)$.

**Adversary's Capabilities.** This work follows the widely used grey-box attack setting in prior studies (Duan et al., 2023; Matsumoto et al., 2023; Kong et al., 2023; Zhai et al., 2024). The grey-box setting is the mainstream attack setting for membership inference attacks against diffusion models. Given that diffusion models generate images through a progressive denoising process, the attacker is assumed to have access to the intermediate outputs produced during generation. This is achieved by specifying the timesteps at which these intermediate outputs are obtained, while access to the internal parameters of the model remains restricted.

**General Paradigm.** Attacks such as Naive (Matsumoto et al., 2023), SecMI (Duan et al., 2023) and PIA (Kong et al., 2023), relying on reconstruction errors, have demonstrated effectiveness in diffusion models. These attacks share common characteristics, utilizing the model's image prediction capability to execute attacks. They can be unified as follows: given the image to be tested $x_i$, calculate the distance between the image $x_{i,t}$ predicted by the model at step $t$ and the target result $x_{i,t}^{target}$ of $x_i$ at step $t$, then using the distance as the membership score. $x_{i,t}$ and $x_{i,t}^{target}$ are obtained in different ways depending on the attack algorithm. Then, the attacker sets a threshold $\tau$. If the score is less than $\tau$, the image is classified as member data; otherwise, it is classified as hold-out data. Formally, these attacks can be formulated as the general paradigm:

$$\mathcal{A}(x_i, \theta) = \mathbb{1} \left[ ||x_{i,t} - x_{i,t}^{target}||_q \leq \tau \right], \tag{6}$$

where $q$ represents the type of norm. *We prove in Appendix B that the error-based attacks can be translated into the general paradigm expressed in Eq. 6.*

### 4.2 FREQUENCY PERSPECTIVE OF MIAs FOR DIFFUSION MODELS

**Frequency Characteristics of Diffusion Models.** Existing attack studies on diffusion models mainly concentrate on reconstruction errors, yet they neglect an important aspect: analyzing models' information processing from the frequency domain. Diffusion models' operational mechanism features distinct frequency hierarchical properties. They first denoise low-frequency signals according to the learned distribution, then utilize specific low-frequency information as prior knowledge to process high-frequency details (Qian et al., 2024). Recovering high-frequency information effectively relies not only on the model's learned distribution but also closely ties to the image's inherent structure and contours during denoising. Previous research (Yang et al., 2023) has shown that diffusion models exhibit more variation and uncertainty in handling high-frequency information.

Current error-based attacks mainly evaluate a model's ability to process individual image. However, high- and low-frequency content varies greatly among images, and diffusion models have different mechanisms for handling such information. These two factors raise an interesting question: Does the high- and low-frequency content within a single image impact existing attack algorithms?

**Frequency Effects on MIAs for Diffusion.** To study how frequency domain information affects MIAs, we visualized the relationship between high-frequency content and existing attacks' membership scores. First, we transform images to the frequency domain using the Fourier transform

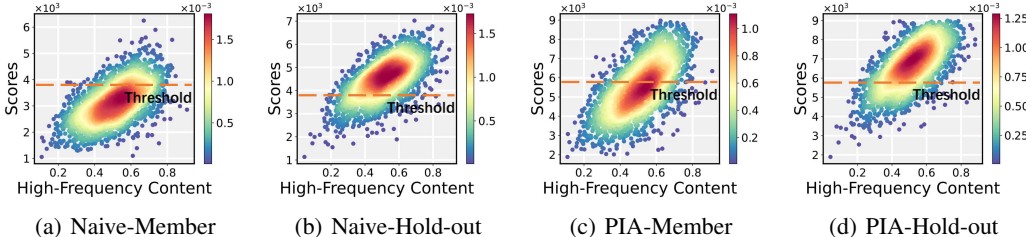

(a) Naive-Member     (b) Naive-Hold-out     (c) PIA-Member     (d) PIA-Hold-out

Figure 1: Statistical plots of membership scores versus high-frequency content for the MS-COCO dataset. Horizontal coordinates indicate high-frequency content and vertical coordinates indicate membership scores. We used red to indicate areas with the highest data density.

and divide the high- and low-frequency regions by setting the frequency domain radius $= 5$ as the high-low frequency boundary. Then, we calculated the percentage of high-frequency content by summing the squared frequency components in both regions. As shown in Fig. 1, visualizing the scores of attacks reveals a trend that as the high-frequency content of the images increased, so did the membership scores. This shows current attacks are biased, giving higher scores to images with more high-frequency content. Higher scores mean a lower degree of fitting, making it more likely to be classified as hold-out data. We term the phenomenon "high-frequency deficiency".

Table 1: High-frequency content statistics for failed samples. Failed samples refer to those that are misclassified, including member data identified as non-member and non-member data identified as member. In the failed samples, the high-frequency content of the member data is significantly larger than that of the hold-out data.

| Method | STLU-10 | | | Tiny-IN | | | MS-COCO | | | Flickr | | |
|---|---|---|---|---|---|---|---|---|---|---|---|---|
| | Member | | Hold-out | Member | | Hold-out | Member | | Hold-out | Member | | Hold-out |
| Naive | 0.598 | > | 0.413 | 0.607 | > | 0.445 | 0.604 | > | 0.424 | 0.612 | > | 0.415 |
| PIA | 0.601 | > | 0.412 | 0.599 | > | 0.402 | 0.616 | > | 0.422 | 0.623 | > | 0.409 |
| SecMI | 0.629 | > | 0.504 | 0.602 | > | 0.419 | 0.632 | > | 0.517 | 0.622 | > | 0.500 |

To dig deeper into the influence of high-frequency deficiency on attacks, we analyzed the high-frequency content of common attack failure cases. As shown in Tab. 1, in the member dataset, images with more high-frequency content are often misclassified as hold-out data. In contrast, in the hold-out dataset, images with less high-frequency content tend to be wrongly labelled as member data. Additionally, we visualized pixel-level distance analysis of images, which helped us accurately assess the contribution of different image components to attack scores. The results show that high-frequency components of images have a greater impact on scores. Specifically, scores fluctuate significantly with changes in high-frequency content, indicating a strong correlation. Due to space limitations, we provide the visualizations in Appendix D.1.

### 4.3 High-Frequency Filter Design for Enhanced Diffusion MIAs

Based on the above observations, the following conclusion can be drawn: existing MIAs have not adequately accounted for the effect of high-frequency deficiency, which leads to their failure in certain specific scenarios. Intuitively, the inherent deficiency of diffusion models in handling high-frequency components leads to confusion in distinguishing between images with different high-frequency content. Further, we explore the effects of high-frequency deficiency from a theoretical perspective. We rely on the previously established definition of membership inference attack capability (Yeom et al., 2018). Definition 1 gives an advantage measure that characterizes how well an algorithm can distinguish between member data and hold-out data.

**Definition 1.** *The membership advantage of algorithm $\mathcal{A}$ is defined as:*

$$Adv^M(\mathcal{A}) = \Pr[\mathcal{A} = 1 | m = 1] - \Pr[\mathcal{A} = 1 | m = 0], \tag{7}$$

*where $\Pr[\mathcal{A} = 1 | m = 1]$ indicates the probability that algorithm $\mathcal{A}$ identifies the data as a member when the data is a member. $\Pr[\mathcal{A} = 1 | m = 0]$ indicates the probability that if the data is hold-out data, algorithm $\mathcal{A}$ will identify it as a member.*

When membership scores follow a normal distribution, Yeom et al. (2018) reinterpret the membership advantage in terms of distributional differences. Specifically, they express the advantage using the ratio between the standard deviation of the hold-out scores $\sigma_H$ and that of the member scores $\sigma_M$. And they further demonstrate that this ratio is positively correlated with the membership advantage:

$$\sigma_H/\sigma_M \propto Adv^M(\mathcal{A}). \tag{8}$$

A larger ratio $\sigma_H/\sigma_M$ corresponds to a greater advantage for the attacker. When $\sigma_H < \sigma_M$, it means that there is no member advantage. Mathematically, the high-frequency deficiency contributes equally to a high standard deviation in the scores of both member and hold-out data, which may lead to a decrease in $\sigma_H/\sigma_M$, weakening the advantages of identifying member data and consequently interfering with the effective differentiation between member and hold-out data. This perspective is further validated in proposition 1.

**Enhanced MIAs Based on High-Frequency Filter.** Having established that high-frequency content introduces variability that masks the membership signal, we propose a simple yet effective solution: selectively filtering out this confounding high-frequency information while preserving the more reliable low-frequency components that carry stronger membership signals. Mathematically, this operation is performed as follows:

$$\mathcal{F}(x_{i,t}) = IFFT(FFT(x_{i,t}) \odot \beta_{i,t}(r)), \tag{9}$$

where $\odot$ denotes element-wise multiplication, and $\beta_{i,t}(r)$ is a mask designed as a filtering factor for frequency, it can be formalized as:

$$\beta_{i,t}(r) = \begin{cases} s & \text{if } r > r_t, \\ 1 & \text{otherwise.} \end{cases} \tag{10}$$

where $s$ serves to implement the frequency-dependent filtering factor, $r$ denotes the frequency domain radius, and $r_t$ is the high-frequency threshold radius. Therefore, our improvement to the general paradigm can be expressed as:

$$\mathcal{A}'(x_i, \theta) = \mathbb{1}\left[||\mathcal{F}(x_{i,t}) - \mathcal{F}(x_{i,t}^{target})||_q \leq \tau\right]. \tag{11}$$

**Proposition 1.** *Assuming the attack has a membership advantage $Adv^M(\mathcal{A})$. Denote the original standard deviations of its membership scores in member and hold-out data as $\sigma_M$ and $\sigma_H$. And the standard deviations after removing the high-frequency components are $\sigma'_M$ and $\sigma'_H$. The standard deviation of membership scores in the high-frequency components is $h_M(h_H)$, and the low-frequency components is $l_M(l_H)$ in member(hold-out) data. Let $l_H - l_M = \Delta$, $h_M = k \cdot h_H$ with $k > 0$. If $k^2 > 1 + \frac{2\Delta}{h_H^2}(l_M + 2\Delta - \sqrt{(l_M + 2\Delta)^2 + h_H^2})$, we have:*

$$\sigma'_H/\sigma'_M > \sigma_H/\sigma_M. \tag{12}$$

*Proof.* We provide the detailed proof of Proposition 1 in Appendix B.

Since $l_M + 2\Delta - \sqrt{(l_M + 2\Delta)^2 + h_H^2} < 0$, $\sigma'_H/\sigma'_M > \sigma_H/\sigma_M$ is hold constantly when $k \geq 1$. The key insight of Proposition 1 is that when the ratio of $h_M/h_H$ exceeds a certain threshold, high-frequency contents will weaken the membership advantage, and filtering the membership scores derived from high-frequency components will amplify the membership advantage. From a theoretical perspective, it demonstrates that high-frequency components reduce the attack capability of the algorithm, while also validating the effectiveness of high-frequency filtering methods.

To further validate the practical applicability of the theoretical framework, we investigated the constraint conditions of $k$. The result shows that $k$ satisfies its constraint conditions under normal circumstances. We also provide a discussion on normality and verify the validity of the normality assumption in Proposition 1, while experimental results further demonstrate that our method does not rely on the assumption of normality. *Detailed experiments and analysis are provided in Appendix D.2.*

## 5 EXPERIMENTS

### 5.1 EXPERIMENT SETUP

**Datasets and Models.** We adhere to the stringent assumption that both member and hold-out data reside within the same distribution. For DDIM, we used the STL10-U (Coates et al., 2011), CIFAR-100 (Krizhevsky et al., 2009), and Tiny-ImageNet(Tiny-IN) (Deng et al., 2009) datasets for training,

Table 2: Under the default training settings, attack performance of baselines in DDIM. High-frequency filter results in a significant improvement in baseline performance.

| Method | STL10-U | | | CIFAR-100 | | | Tiny-IN | | |
|---|---|---|---|---|---|---|---|---|---|
| | ASR | AUC | TPR@1%FPR | ASR | AUC | TPR@1%FPR | ASR | AUC | TPR@1%FPR |
| Naive | 73.60 | 79.60 | 5.96 | 70.41 | 77.01 | 7.13 | 74.60 | 81.97 | 7.96 |
| **Naive+F** | 77.98 | 84.40 | 7.50 | 75.01 | 82.36 | 9.73 | 80.69 | 87.60 | 13.29 |
| SecMI | 81.14 | 87.39 | 11.11 | 80.56 | 87.21 | 16.50 | 82.91 | 89.60 | 13.96 |
| **SecMI+F** | 86.51 | 91.39 | 14.63 | 88.09 | 93.74 | 24.32 | 90.31 | 93.82 | 25.79 |
| PIA | 80.43 | 87.45 | 9.98 | 77.51 | 84.80 | 12.27 | 80.87 | 86.30 | 14.66 |
| **PIA+F** | 86.81 | 92.11 | 19.57 | 85.05 | 92.20 | 23.34 | 89.12 | 93.23 | 32.91 |
| **Avg+** | **+5.38** | **+4.49** | **+4.48** | **+6.56** | **+6.43** | **+7.16** | **+7.25** | **+5.59** | **+11.80** |

respectively. Specifically, we randomly selected 50% of the training set to serve as member data, while the remaining 50% was designated as hold-out data. For text-to-image diffusion models, we employed 416/417 samples on Pokémon (Lambda, 2023), 2500/2500 samples on MS-COCO (Lin et al., 2014), and 1000/1000 samples on Flickr (Young et al., 2014) as the member/hold-out dataset, utilizing stable diffusion v1-4 (CompVis, 2024) for fine-tuning. Furthermore, for pre-trained diffusion models, we selected stable diffusion v1-4 and v1-5 (RunwayML, 2024) as attack targets. We adhere to the settings in (Dubiński et al., 2024; Zhai et al., 2024), employing Laion-MI (Dubiński et al., 2024) dataset to ensure that both member and hold-out data exhibit the same distribution. *Detailed training information is provided in Appendix D.3.*

**Evaluation Metrics.** We use the established metrics employed in prior research (Matsumoto et al., 2023; Duan et al., 2023; Kong et al., 2023), which include the Attack Success Rate (ASR), the Area Under the Curve (AUC), and the True Positive Rate (TPR) at 1% False Positive Rate (FPR) (denoted as TPR@1%FPR).

**Baselines.** We employed the SecMI (Duan et al., 2023), PIA (Kong et al., 2023), and Naive (Matsumoto et al., 2023) as our baselines for comparison. These approaches are characterized as error-based attacks and operate effectively within a grey-box setting. We adopt the parameters advised in their respective publications. Since our method is a unified improvement of previous attacks, we follow the threshold acquisition procedure described in the publications of these baselines.

**Implementation Details.** Both training (fine-tuning) and inference are conducted on a single RTX 3090 GPU(24G). We set $s = 0.2$, $r_t = 5$ in our method and use $\ell_2$ norm under the general paradigm.

Table 3: Under the default training settings, attack performance in fine-tuned stable diffusion.

| Method | Pokémon | | | MS-COCO | | | Flickr | | |
|---|---|---|---|---|---|---|---|---|---|
| | ASR | AUC | TPR@1%FPR | ASR | AUC | TPR@1%FPR | ASR | AUC | TPR@1%FPR |
| Naive | 79.50 | 86.97 | 6.49 | 80.29 | 87.85 | 4.80 | 79.29 | 86.14 | 16.59 |
| **Naive+F** | 87.88 | 94.14 | 41.25 | 93.60 | 98.32 | 41.99 | 90.90 | 96.82 | 67.60 |
| SecMI | 76.37 | 83.16 | 12.74 | 82.09 | 89.37 | 16.79 | 71.49 | 77.31 | 6.19 |
| **SecMI+F** | 83.75 | 89.73 | 31.25 | 91.00 | 95.74 | 27.40 | 80.10 | 85.95 | 21.20 |
| PIA | 72.27 | 76.76 | 7.75 | 68.19 | 72.88 | 5.20 | 64.60 | 67.95 | 5.79 |
| **PIA+F** | 80.87 | 85.44 | 39.25 | 76.00 | 83.08 | 16.59 | 69.30 | 74.62 | 19.60 |
| **Avg+** | **+8.12** | **+7.47** | **+28.26** | **+10.01** | **+9.01** | **+19.73** | **+8.31** | **+8.66** | **+26.61** |

## 5.2 OVERALL PERFORMANCE

**Denoising Diffusion Implicit Models.** For DDIM, we compared the performance of all baselines before and after adding the high-frequency filter, with the relevant results detailed in Tab. 2. We evaluated the average performance improvements of various baselines in different metrics. The experimental results clearly indicate that the filter significantly improves the performance of all baselines. Moreover, the higher the complexity of dataset, the more pronounced this performance improvement becomes. Taking Tiny-IN as an example, after adding the filter, the ASR and AUC improved by 7.25% and 5.59% on average. The increase in the TPR@1% FPR metric was even more significant, with an average improvement of 11.80% and a maximum improvement of 18.25%.

**Stable Diffusion Models.** The experimental results for the fine-tuned stable diffusion attacks are presented in Tab. 3. Based on the analysis of the average improvement over baselines, we observed that the ASR improved by 10.01%, AUC improved by 9.01%, and the improvement in the TPR@1% FPR reached as high as 19.73% on the MS-COCO dataset. Notably, on Flickr, our method achieved the highest TPR@1% FPR improvement of 51.01% in Naive. These results strongly indicate that our method significantly enhances the attack efficacy of the baselines across diverse data distributions and scales by mitigating high-frequency deficiency. We emphasize the generality of our proposed method and validate its effectiveness in other classes of attack methods in Appendix D.5. Moreover, we have conducted tests on the pre-trained stable diffusion, and the results indicate that the filter exhibits only a modest effect, possibly due to the inherent shortcomings of the baselines. *A detailed discussion is provided in Appendix D.4.*

## 5.3 IN-DEPTH ANALYSIS OF ATTACK PERFORMANCE

We have theoretically proven that removing high-frequency deficiency can amplify the distinction between member and hold-out data. To further validate our conjectures, we visualize the membership scores of the baselines before and after applying the filter. As illustrated in Fig. 2, it is evident that the distribution gap between member data and hold-out data has increased noticeably after applying our method. Blue boxes mark the areas where member and hold-out data interleave, indicating member/hold-out data are indistinguishable by thresholds. After applying the filter, we observe a significant reduction in sample interleavements. This compelling evidence validates our conjectures and demonstrates the filter's effectiveness. *More visualizations will be presented in Appendix D.6.*

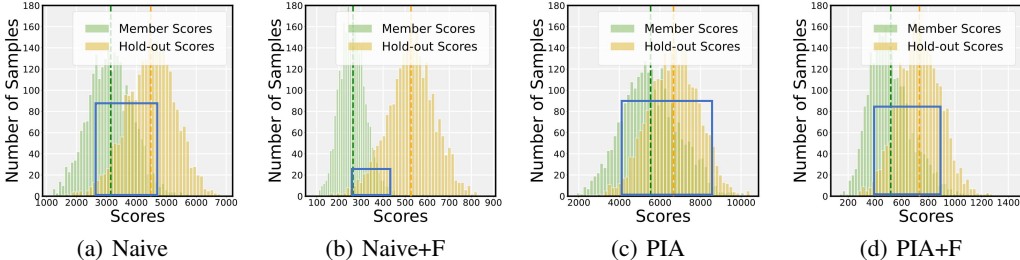

|   (a) Naive   |   (b) Naive+F   |   (c) PIA   |   (d) PIA+F   |

Figure 2: Membership score distribution of member and hold-out data in the MS-COCO dataset. The score distribution gap between member data and hold-out data has noticeably increased.

Table 4: Naive'attack performance at different $s$ and $r_t$ in MS-COCO dataset. Our method has lower sensitivity to hyperparameters and performs excellently under most parameter settings.

| $s/r_t$ | $s = 0.0$ | | $s = 0.1$ | | $s = 0.2$ | | $s = 0.3$ | | $s = 0.4$ | | $s = 0.5$ | |
|---|---|---|---|---|---|---|---|---|---|---|---|---|
| | ASR | AUC | ASR | AUC | ASR | AUC | ASR | AUC | ASR | AUC | ASR | AUC |
| $r_t = 1$ | 69.40 | 73.93 | 81.49 | 89.15 | 82.99 | 90.24 | 80.80 | 88.12 | 80.50 | 87.87 | 80.30 | 87.48 |
| $r_t = 3$ | 93.70 | 97.63 | 93.59 | 97.59 | 92.69 | 96.47 | 89.20 | 94.22 | 84.79 | 91.93 | 83.39 | 90.04 |
| $r_t = 5$ | 94.10 | 97.89 | 94.59 | 97.85 | 93.60 | 98.32 | 90.60 | 95.23 | 87.59 | 93.61 | 84.89 | 91.50 |
| $r_t = 7$ | 93.60 | 97.27 | 93.40 | 96.88 | 91.39 | 96.50 | 91.20 | 95.75 | 88.80 | 94.13 | 86.40 | 92.49 |
| $r_t = 9$ | 92.00 | 96.48 | 91.69 | 96.61 | 91.00 | 95.71 | 90.60 | 95.13 | 88.20 | 94.02 | 86.90 | 92.58 |
| $r_t = 10$ | 91.90 | 96.30 | 91.80 | 96.04 | 90.70 | 95.68 | 89.99 | 94.78 | 87.80 | 93.91 | 87.00 | 92.57 |

## 5.4 ABLATION STUDY

To investigate the impact of the filter under different hyperparameter settings, we adjusted the high-frequency threshold $r_t$ and the filtering parameter $s$. Experiments were conducted with various values for $r_t$ and $s$ on the MS-COCO dataset using Naive attack, and the results are presented in Tab. 4. From the experimental analysis, we recommend a value range for $r_t$ of $[3, 10]$ and for $s$ of $[0.0, 0.3]$. Within this range, the filter achieves optimal performance, significantly enhancing the baseline performance while exhibiting low sensitivity to changes in hyperparameters, further demonstrating the robustness of the filter. When $r_t = 1$, the high-frequency threshold is set too low, leading to most frequency components of the image being filtered. This contradicts our intention to only suppress high-frequency deficiency, resulting in a significant decline in attack performance.

When $s = 0.4$ and $s = 0.5$, the filtering effect on the high-frequency components is weakened. Although the baseline performance improves significantly, it does not reach the optimal state. *More ablation experiments will be presented in Appendix D.8.*

## 5.5 MORE STRINGENT ATTACK CONDITIONS

**Weaker Overfitting.** When fine-tuning stable diffusion, the number of iterations is typically determined by the user's specific needs. In some scenarios, extensive iterations are not required to fine-tune the model. To simulate a model with a lower degree of overfitting, we halved the number of iterations across all datasets, as referenced in Zhai et al. (2024). As shown in Tab. 5, the filter demonstrates nice effectiveness. Taking the Flickr dataset as an example, after applying the filter, the ASR improved by an average of 4.34%, with a maximum improvement of 9.25% in Naive. The AUC increased by an average of 7.04%, with the Naive demonstrating the most significant improvement of 14.53%. Compared to the default settings, the filter's effect is diminished, which is directly linked to the performance decline of the baseline under the weaker overfitting. Therefore, the results can still prove the effectiveness of the high-frequency filter.

Table 5: Attack performance with weaker overfitting assumption in fine-tuned stable diffusion.

| Method | Pokémon | | | MS-COCO | | | Flickr | | |
|---|---|---|---|---|---|---|---|---|---|
| | ASR | AUC | TPR@1%FPR | ASR | AUC | TPR@1%FPR | ASR | AUC | TPR@1%FPR |
| Naive | 70.23 | 74.85 | 5.00 | 60.79 | 63.51 | 1.60 | 67.50 | 68.65 | 5.49 |
| **Naive+F** | 79.41 | 84.98 | 12.60 | 70.30 | 75.43 | 5.40 | 76.75 | 83.18 | 14.00 |
| SecMI | 70.31 | 74.46 | 4.25 | 67.19 | 71.99 | 3.60 | 71.24 | 77.09 | 4.59 |
| **SecMI+F** | 72.26 | 76.44 | 5.80 | 69.10 | 73.79 | 4.79 | 72.25 | 79.09 | 6.00 |
| PIA | 66.39 | 67.77 | 3.50 | 55.09 | 53.64 | 0.80 | 60.00 | 59.76 | 2.90 |
| **PIA+F** | 68.71 | 71.89 | 5.20 | 57.04 | 57.10 | 2.20 | 62.25 | 64.35 | 3.50 |
| **Avg+** | **+4.48** | **+5.41** | **+3.61** | **+4.46** | **+5.73** | **+2.13** | **+4.34** | **+7.04** | **+3.51** |

**Without Captions.** We investigated the effectiveness of our method when the attacker has no access to the corresponding text of the images. The results demonstrate that our method remains effective under this condition. *Detailed experiments and results are provided in the Appendix D.7.*

## 5.6 IMPACT OF DEFENSE

To investigate the impact of defense mechanisms on our method, we examine two defenses against membership inference attacks (MIAs): the commonly used data augmentation strategy and the state-of-the-art memory mitigation technique $SS_{e_i}$ (Wen et al., 2024). Recall that MIAs primarily benefit from overfitting. Data augmentation techniques are typically employed to prevent overfitting. During the fine-tuning process of stable diffusion, it employs Random-Crop and Random-Flip by default (Face, 2024). Memory mitigation dynamically evaluates the model's memory of the data during training and adjusts the training process accordingly, thereby significantly reducing overfitting and training data leakage. Both serve as effective defenses against MIAs. As shown in Tab. 6, the performance of the baseline methods gradually decreases with the inclusion of defenses. Nevertheless, our method continues to exhibit strong performance, further confirming its robustness in the presence of such defenses. *More detailed results are provided in the Appendix D.11.*

Table 6: Attack performance AUC under the defenses. With the introduction of defense mechanisms, the baselines exhibit varying degrees of performance degradation. Our method remains effective and continues to deliver a clear performance improvement over the baselines under these defenses.

| $SS_{e_i}$ | DataAug | Naive | Naive+F | Gain | SecMI | SecMI+F | Gain | PIA | PIA+F | Gain |
|---|---|---|---|---|---|---|---|---|---|---|
| × | × | 87.95 | 96.81 | +8.86 | 86.67 | 91.42 | +4.75 | 82.64 | 92.54 | +9.90 |
| × | ✓ | 86.87 | 94.14 | +7.27 | 83.16 | 89.73 | +6.57 | 76.76 | 85.44 | +8.86 |
| ✓ | × | 70.23 | 76.91 | +6.68 | 81.00 | 86.82 | +5.82 | 68.81 | 76.32 | +7.51 |
| ✓ | ✓ | 69.09 | 75.23 | +6.14 | 79.32 | 84.42 | +5.10 | 61.09 | 67.95 | +6.86 |

**Adaptive Defense.** We propose an adaptive defense strategy that applies random amplitude suppression, reducing the low-frequency components of images to 70-90% of their original values, and adds noise with a variance of $0.05$ to the low-frequency domain during the fine-tuning phase. The core mechanism of this strategy lies in moderately suppressing the representation weights of low-frequency components, thereby reducing the model's overfitting to low-frequency information in images. We conduct experiments using the Pokémon dataset. As shown in Tab. 7, after applying the adaptive defense, the baseline attack performance is reduced to some extent, and the gain achieved by our method is also mitigated. But, this operation may lead to a decline in model performance, as it introduces perturbations in the low-frequency components of the images during training.

Table 7: Attack performance under adaptive defense in Pokémon dataset.

| Method | ASR | AUC | TPR@1%FPR |
|---|---|---|---|
| Naive | 74.34 | 81.12 | 5.06 |
| **Naive+F** | 76.66 | 83.59 | 8.24 |
| SecMI | 73.13 | 78.59 | 3.37 |
| **SecMI+F** | 77.47 | 84.78 | 13.25 |
| PIA | 68.43 | 72.96 | 6.67 |
| **PIA+F** | 74.66 | 78.96 | 14.63 |
| **Avg+** | **+3.96** | **+4.89** | **+7.01** |

## 6 CONCLUSION

In this paper, we define a general paradigm for the error-based MIAs for diffusion models. Under this general paradigm, we find that the current attacks ignore the intrinsic deficiency of the diffusion model in handling the high-frequency components, which results in limited attack performance. To address this, we introduce a simple and efficient method which mitigates the negative impact of high-frequency deficiency on MIAs by filtering images' high-frequency information. Experimental results reveal that our method can be seamlessly incorporated into attacks within the general paradigm, significantly enhancing attack performance across diverse settings.

**Limitations.** Our method exhibits limited efficacy in the pre-training setting, likely due to the overall substandard performance of existing attacks in this setting. This issue may arise from the fact that the pre-trained model does not align with the current assumptions related to overfitting. Future investigations need to thoroughly explore MIAs in the pre-training setting.

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

# A  MORE DETAILS FOR RELATED WORK

## A.1  DENOISING DIFFUSION PROBABILISTIC MODEL (DDPM)

DDPM (Ho et al., 2020) consists of a forward process and a denoising process. In the forward process, DDPM transitions from an intractable data distribution, represented as $x_0 \sim q_0(x_0)$, to a Gaussian distribution $q_T(x_T) \sim \mathcal{N}(x_T; 0, I)$. This transition is achieved by progressively adding Gaussian noise to the original image $x_0$. Consequently, the transition distribution at timestep $t$ is defined as follows:

$$q(x_t|x_{t-1}) = \mathcal{N}\left(x_t; \sqrt{\alpha_t}x_{t-1}, (1-\alpha_t)I\right), \tag{A.1}$$

where $\alpha_1, \ldots, \alpha_T$ are the predefined noise schedules. Leveraging the properties of chained Gaussian processes, DDPM defines $\overline{\alpha}_t = \prod_{i=1}^{t} \alpha_i$. Consequently, the value of $x_t$ is computed in a single step:

$$x_t = \sqrt{\overline{\alpha}_t}x_0 + \sqrt{1-\overline{\alpha}_t}\epsilon, \quad \epsilon \sim \mathcal{N}(0, I). \tag{A.2}$$

DDPM allows for a simplification of the optimization objective:

$$L_{sample} = \mathbb{E}_{x_0,\epsilon}\left[\|\epsilon - \epsilon_\theta(x_t, t)\|^2\right], \tag{A.3}$$

where $\epsilon_\theta(x_t, t)$ is predicted by the diffusion models. The denoising (or reverse) process shares the same functional form as the forward process (Sohl-Dickstein et al., 2015). It is expressed as a Gaussian transition characterized by a learned mean and a fixed variance:

$$p_\theta(x_{t-1}|x_t) = \mathcal{N}(x_{t-1}; \frac{1}{\sqrt{\alpha_t}}\left(x_t - \frac{1-\alpha_t}{\sqrt{1-\overline{\alpha}_t}}\epsilon_\theta(x_t, t)\right), \sigma_t^2 I). \tag{A.4}$$

The denoising process defined by DDPM is outlined as follows:

$$x_{t-1} = \frac{1}{\sqrt{\alpha_t}}\left(x_t - \frac{1-\alpha_t}{\sqrt{1-\overline{\alpha}_t}}\epsilon_\theta(x_t, t)\right) + \sigma_t z, \tag{A.5}$$

where $z \sim \mathcal{N}(0, I)$, which brings uncertainty and diversity to the denoising process.

## A.2  DEFENSE AGAINST MODEL MEMORIZATION.

Wen et al. (2024) identified that when a model exactly memorizes training data, the noise prediction network exhibits a pronounced discrepancy between its conditional and unconditional predictions. Given a training data $x$, and the caption embedding $e$ consisting of $N$ tokens, they formulate the minimization objective at step $t$ as:

$$\mathcal{L}(x_t, e) = \|\epsilon_\theta(x_t, e) - \epsilon_\theta(x_t, \varnothing)\|_2, \tag{A.6}$$

where $\epsilon_\theta$ denotes the noise predictor and $\varnothing$ is the null prompt. The memorization score for each token at position $i \in [0, N-1]$ is then defined as:

$$SS_{e_i} = \frac{1}{T}\sum_{t=1}^{T}\|\nabla_{e_i}\mathcal{L}(x_t, e)\|_2. \tag{A.7}$$

To mitigate memorization, they propose excluding a sample from the mini-batch whenever this score exceeds a predefined threshold, thereby skipping the loss computation for that sample. Since such samples have already been seen by the model during training, their removal is unlikely to degrade overall model performance. This method is proven to significantly alleviate exact memorization of training samples, providing protection for the privacy of the training set.

# B  GENERAL PARADIGM

Naive (Matsumoto et al., 2023) determines member data through the loss function, specifically judging based on the distance between the added noise and the predicted noise. SecMI (Duan et al., 2023) leverages diffusion inversion to obtain the ground truth at step $t$ and step $t-1$. Based on this, the model predicts $x_{t-1}$ from the ground truth at step $t$ and assesses whether a sample is a member

by examining the distance between the predicted $x_{t-1}$ and its ground truth. PIA (Kong et al., 2023) retrieves an initial noise through a proximal initialization process, then uses the model to predict noise for samples containing that initial noise. Finally, it evaluates the membership status based on the distance between the predicted noise and the initial noise. In this section, we will demonstrate that the baselines Naive, PIA and SecMI can be translated into the general paradigm we have defined, and here we use the $\ell_1$ norm as an example.

**Naive:** According to Eq. A.2, $x_0$ can be expressed as:

$$x_0 = \frac{x_t - \sqrt{1 - \bar{\alpha}_t}\,\epsilon}{\sqrt{\bar{\alpha}_t}}, \epsilon \sim \mathcal{N}(0, I). \tag{B.1}$$

Naive recognizes membership based on the following inequality:

$$||\epsilon - \epsilon_\theta \left( \sqrt{\bar{\alpha}_t}x_0 + \sqrt{1 - \bar{\alpha}_t}\epsilon\,, t \right)|| < c, \tag{B.2}$$

where $\epsilon_\theta(\cdot)$ indicates the predicted noise. Multiply $\sqrt{1 - \bar{\alpha}_t}$ on both sides of Eq. B.2:

$$||\sqrt{1 - \bar{\alpha}_t}\epsilon - \sqrt{1 - \bar{\alpha}_t}\epsilon_\theta \left( \sqrt{\bar{\alpha}_t}x_0 + \sqrt{1 - \bar{\alpha}_t}\epsilon\,, t \right)|| < \sqrt{1 - \bar{\alpha}_t}c, \tag{B.3}$$

which is equivalent to:

$$||\sqrt{1 - \bar{\alpha}_t}\epsilon - x_t + x_t - \sqrt{1 - \bar{\alpha}_t}\epsilon_\theta \left( \sqrt{\bar{\alpha}_t}x_0 + \sqrt{1 - \bar{\alpha}_t}\epsilon\,, t \right)|| < \sqrt{1 - \bar{\alpha}_t}c. \tag{B.4}$$

Divide both sides of Eq. B.4 by $\sqrt{\bar{\alpha}_t}$:

$$||\frac{\sqrt{1 - \bar{\alpha}_t}\epsilon - x_t + x_t - \sqrt{1 - \bar{\alpha}_t}\epsilon_\theta \left( \sqrt{\bar{\alpha}_t}x_0 + \sqrt{1 - \bar{\alpha}_t}\epsilon\,, t \right)}{\sqrt{\bar{\alpha}_t}}|| < \frac{\sqrt{1 - \bar{\alpha}_t}c}{\sqrt{\bar{\alpha}_t}}. \tag{B.5}$$

According to Eq. B.1, we can set:

$$x_0^{target} = \frac{x_t - \sqrt{1 - \bar{\alpha}_t}\epsilon}{\sqrt{\bar{\alpha}_t}}, x_0 = \frac{x_t - \sqrt{1 - \bar{\alpha}_t}\epsilon_\theta \left( \sqrt{\bar{\alpha}_t}x_0 + \sqrt{1 - \bar{\alpha}_t}\epsilon\,, t \right)}{\sqrt{\bar{\alpha}_t}}. \tag{B.6}$$

Therefore, Eq. B.5 can be converted to the distance between the original image and the target image:

$$||x_0 - x_0^{target}|| \leq \tau, \tag{B.7}$$

where $\tau = c\sqrt{1 - \bar{\alpha}_t}/\sqrt{\bar{\alpha}_t}$.

**PIA:** In order to reduce the error caused by random noise, the authors used proximal initialization to obtain the initial noise so as to improve Naive. It is expressed as:

$$||\epsilon_\theta(x_0, 0) - \epsilon_\theta \left( \sqrt{\bar{\alpha}_t}x_0 + \sqrt{1 - \bar{\alpha}_t}\epsilon_\theta(x_0, 0), t \right)|| < c, \tag{B.8}$$

where $\epsilon_\theta(x_0, 0)$ represents the noise prediction for $x_0$. In the same way, Eq. B.8 can be converted to:

$$||\frac{\sqrt{1 - \bar{\alpha}_t}\epsilon(x_0, 0) - x_t + x_t - \sqrt{1 - \bar{\alpha}_t}\epsilon_\theta \left( \sqrt{\bar{\alpha}_t}x_0 + \sqrt{1 - \bar{\alpha}_t}\epsilon(x_0, 0)\,, t \right)}{\sqrt{\bar{\alpha}_t}}|| < \frac{\sqrt{1 - \bar{\alpha}_t}c}{\sqrt{\bar{\alpha}_t}}. \tag{B.9}$$

According to Eq. B.1, we can set:

$$x_0^{target} = \frac{x_t - \sqrt{1 - \bar{\alpha}_t}\epsilon_\theta(x_0, 0)}{\sqrt{\bar{\alpha}_t}}, x_0 = \frac{x_t - \sqrt{1 - \bar{\alpha}_t}\epsilon_\theta \left( \sqrt{\bar{\alpha}_t}x_0 + \sqrt{1 - \bar{\alpha}_t}\epsilon_\theta(x_0, 0)\,, t \right)}{\sqrt{\bar{\alpha}_t}}. \tag{B.10}$$

Therefore, Eq. B.9 can be converted to the distance between the original image and the target image:

$$||x_0 - x_0^{target}|| \leq \tau, \tag{B.11}$$

where $\tau = c\sqrt{1 - \bar{\alpha}_t}/\sqrt{\bar{\alpha}_t}$.

**SecMI:** Inspired by recent works on deterministic reversing and sampling from diffusion models (Song et al., 2020b; Kim et al., 2022), SecMI used DDIM and DDIM inversion deterministic sampling in the forward and backward processes for the samples to be tested:

$$x_{t+1} = \phi_\theta(x_t, t) = \sqrt{\bar{\alpha}_{t+1}}\frac{x_t - \sqrt{1 - \bar{\alpha}_t}\epsilon_\theta(x_t, t)}{\sqrt{\bar{\alpha}_t}} + \sqrt{1 - \bar{\alpha}_{t+1}}\epsilon_\theta(x_t, t), \tag{B.12}$$

$$x_{t-1} = \psi_\theta(x_t, t) = \sqrt{\bar{\alpha}_{t-1}} \frac{x_t - \sqrt{1 - \bar{\alpha}_t} \epsilon_\theta(x_t, t)}{\sqrt{\bar{\alpha}_t}} + \sqrt{1 - \bar{\alpha}_{t-1}} \epsilon_\theta(x_t, t). \quad \text{(B.13)}$$

Denote by $\boldsymbol{\Phi}_\theta(x_s, t)$ the deterministic reverse, i.e., from $x_s$ to $x_t$ ($s < t$):

$$x_t = \boldsymbol{\Phi}_\theta(x_s, t) = \phi_\theta(\cdots \phi_\theta(\phi_\theta(x_s, s), s+1), t-1), \quad \text{(B.14)}$$

and $\boldsymbol{\Psi}_\theta(x_t, s)$ the deterministic denoise process, i.e., from $x_t$ to $x_s$:

$$x_s = \boldsymbol{\Psi}_\theta(x_t, s) = \psi_\theta(\cdots \psi_\theta(\psi_\theta(x_t, t), t-1), s+1). \quad \text{(B.15)}$$

Then, SecMI define $t$-error as the approximated posterior estimation error at step $t$, The algorithm is defined as:

$$\|\psi_\theta(\phi_\theta((\tilde{x}_t, t), t) - \tilde{x}_t\| < c, \quad \text{(B.16)}$$

where $\tilde{x}_t = \boldsymbol{\Phi}_\theta(x_0, t)$. A series of transformations are designed to fetch $x_{t+1}^{target}$ and $x_t^{target}$ and predict $x_t$ by $x_{t+1}^{target}$. Therefore, Eq. B.16 can be expressed as:

$$||x_t - x_t^{target}|| \leq \tau, \quad \text{(B.17)}$$

where $\tau = c$, $x_t = \psi_\theta(\phi_\theta((\tilde{x}_t, t), t))$, $x_t^{target} = \tilde{x}_t$.

## C  PROOF OF PROPOSITION 1

**Proposition 1.** *Assuming the attack has a membership advantage $Adv^M(\mathcal{A})$. Denote the original standard deviations of its membership scores in member and hold-out data as $\sigma_M$ and $\sigma_H$. And the standard deviations after removing the high-frequency components are $\sigma'_M$ and $\sigma'_H$. The standard deviation of membership scores in the high-frequency components is $h_M(h_H)$, and the low-frequency components is $l_M(l_H)$ in member(hold-out) data. Let $l_H - l_M = \Delta$, $h_M = k \cdot h_H$ with $k > 0$. If $k^2 > 1 + \frac{2\Delta}{h_H^2}(l_M + 2\Delta - \sqrt{(l_M + 2\Delta)^2 + h_H^2})$, we have:*

$$\sigma'_H / \sigma'_M > \sigma_H / \sigma_M. \quad \text{(C.1)}$$

*Proof.* The standard deviation before and after filtering high-frequency information satisfies $\sigma_H > \sigma_M$ and $\sigma'_H > \sigma'_M$. $\sigma'_H / \sigma'_M > \sigma_H / \sigma_M$ is equivalent to:

$$\sigma'_H \sigma_M > \sigma_H \sigma'_M. \quad \text{(C.2)}$$

Let $\sigma_H = \sigma'_H + \Delta_H$ and $\sigma_M = \sigma'_M + \Delta_M$. Eq. C.2 is equivalent to:

$$\sigma'_H \sigma'_M + \sigma'_H \Delta_M > \sigma'_H \sigma'_M + \Delta_H \sigma'_M. \quad \text{(C.3)}$$

If we have:

$$k^2 > 1 + \frac{2\Delta}{h_H^2}(l_M + 2\Delta - \sqrt{(l_M + 2\Delta)^2 + h_H^2}). \quad \text{(C.4)}$$

Eq. C.4 can be expressed as:

$$(l_M + \Delta)^2 + k^2 h_H^2 > (l_M + 2\Delta)^2 + h_H^2 - 2\Delta\sqrt{(l_M + 2\Delta)^2 + h_H^2} + \Delta^2. \quad \text{(C.5)}$$

Let $t = (l_M + \Delta)^2 + k^2 h_H^2$, we have:

$$\sqrt{t} > \sqrt{(l_M + 2\Delta)^2 + h_H^2} - \Delta. \quad \text{(C.6)}$$

Therefore, we have:

$$(\sqrt{t} + \Delta)^2 > (l_M + 2\Delta)^2 + h_H^2, \quad \text{(C.7)}$$

which is equivalent to:

$$t > 2l_M\Delta + 2\Delta^2 - 2\Delta\sqrt{t} + (l_M + \Delta)^2 + h_H^2, \quad \text{(C.8)}$$

Substitute back $t = (l_M + \Delta)^2 + k^2 h_H^2$ to Eq. C.8, we obtain:

$$(l_M + \Delta)^2 + k^2 h_H^2 > 2l_M\Delta + 2\Delta^2 - 2\Delta\sqrt{(l_M + \Delta)^2 + k^2 h_H^2} + (l_M + \Delta)^2 + h_H^2. \quad \text{(C.9)}$$

which is equivalent to:

$$(k^2 - 1)h_H^2 > 2l_M\Delta + 2\Delta^2 - 2\Delta\sqrt{(l_M + \Delta)^2 + k^2 h_H^2}. \tag{C.10}$$

Substitute $k = h_M/h_H$ into Eq. C.10:

$$l_M^2 + h_H^2 > (l_M + \Delta)^2 + h_H^2 - 2\Delta\sqrt{(l_M + \Delta)^2 + h_M^2} + \Delta^2. \tag{C.11}$$

Then, we obtain:

$$\sqrt{l_M^2 + h_M^2} > \sqrt{(l_M + \Delta)^2 + h_H^2} - \Delta. \tag{C.12}$$

Since $l_H - l_M = \Delta = \sigma'_H - \sigma'_M > 0$, we obtain:

$$\sqrt{l_M^2 + h_M^2} - \sqrt{l_H^2 + h_H^2} > l_M - l_H. \tag{C.13}$$

Since $\sigma_H = \sigma'_H + \Delta_H$ and $\sigma_M = \sigma'_M + \Delta_M$, the errors of the high- and low-frequency are independent. From the principle of normal distribution superposition, this gives:

$$\Delta_M = \sqrt{l_M^2 + h_M^2} - \sqrt{l_M^2}, \Delta_H = \sqrt{l_H^2 + h_H^2} - \sqrt{l_H^2}. \tag{C.14}$$

Therefore, we have:

$$\Delta_M - \Delta_H > 0. \tag{C.15}$$

So, we have:

$$\sigma'_H\sigma'_M + \sigma'_H\Delta_M > \sigma'_H\sigma'_M + \Delta_H\sigma'_M, \tag{C.16}$$

which is equivalent to:

$$\sigma'_H/\sigma'_M > \sigma_H/\sigma_M. \tag{C.17}$$

Therefore, Proposition 1 is proved.

## D    COMPLEMENTARY EXPERIMENTS

### D.1    MORE FREQUENCY DOMAIN ANALYSIS

As shown in Fig. 3, on the Flickr dataset, membership scores statistics align with our conjecture: they are positively correlated with high-frequency content, increasing as the latter rises.

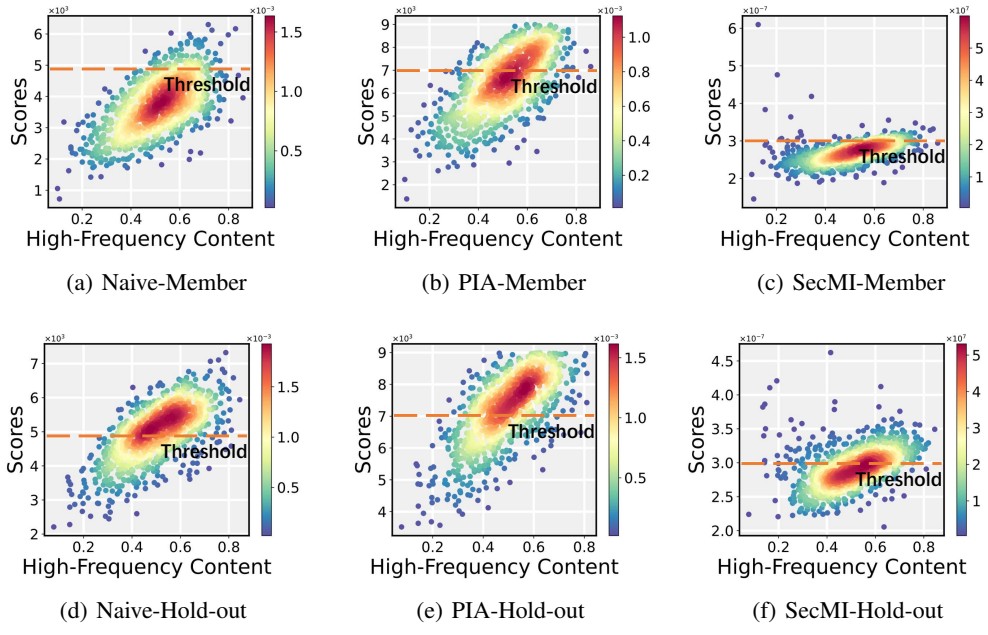

Figure 3: Statistical plots of membership scores versus high-frequency content on the Flickr dataset.

In addition, we visualize the distribution of membership scores contributions in Fig. 4 and Fig. 5, comparing the original images with the pixel-wise errors. Color depth characterizes the magnitude of errors; the deeper the color, the larger the corresponding error at that location. We have observed that areas of high error often coincide with areas of high-frequency information. Due to the variability in high-frequency content across different images, the extent of errors displays significant differences.

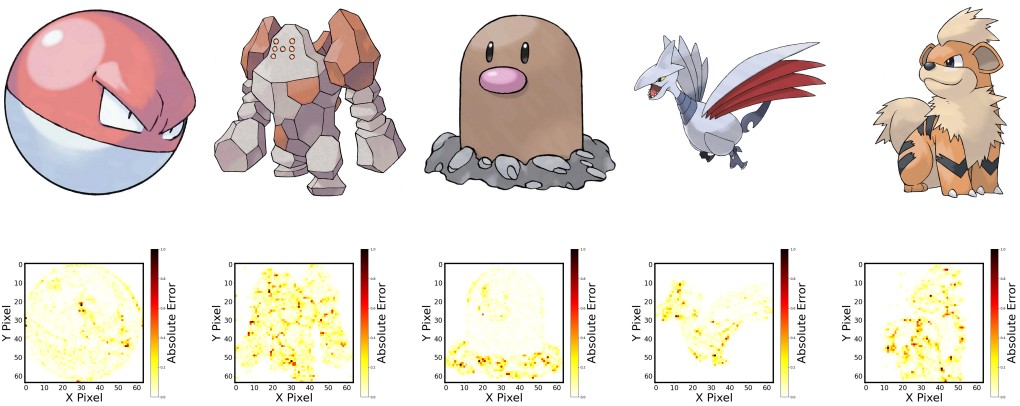

Figure 4: Naive pixel-wise errors distribution visualization, with the top half being the original image and the bottom half being the error visualization. The areas of high error often coincide with areas of high-frequency information.

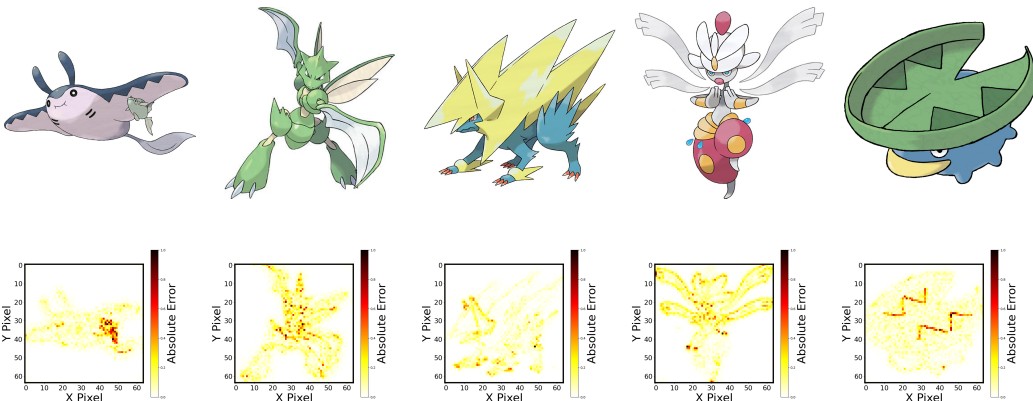

Figure 5: PIA pixel-wise errors distribution visualization. The areas of high error often coincide with areas of high-frequency information.

### D.2 DISCUSSION OF THE THEORETICAL ANALYSIS

**Constraint Conditions.** We conducted experimental validation of the constraint in Proposition 1. As shown in Tab. 8, the parameter $k$ satisfies the constraint across different datasets and models, and in many cases, it even holds that $k > 1$. This further demonstrates the practical feasibility and effectiveness of our theoretical results.

Table 8: Let $f = 1 + \frac{2\Delta}{h_H^2}(l_M + 2\Delta - \sqrt{(l_M + 2\Delta)^2 + h_H^2})$, we compared the values of $k^2$ and $f$ in different datasets and methods, $k$ always satisfies its constraint conditions.

| Method | Tiny-IN | | | STLU-10 | | | MS-COCO | | | Flickr | | |
|---|---|---|---|---|---|---|---|---|---|---|---|---|
| | $k^2$ | | $f$ | $k^2$ | | $f$ | $k^2$ | | $f$ | $k^2$ | | $f$ |
| Naive | 0.988 | > | 0.921 | 0.976 | > | 0.932 | 1.139 | > | 0.924 | 0.910 | > | 0.892 |
| PIA | 0.976 | > | 0.927 | 1.290 | > | 0.904 | 1.290 | > | 0.914 | 1.317 | > | 0.909 |
| SecMI | 1.102 | > | 0.920 | 0.959 | > | 0.897 | 1.778 | > | 0.919 | 1.102 | > | 0.911 |

**Normality Test.** The visualization of membership score distributions in Fig. 6 shows that the score distributions for member and hold-out data exhibit an approximately symmetric unimodal characteristic, aligning with the fundamental traits of a normal distribution (centrality and symmetry). To further ascertain the conformity of the scores to a normal distribution, we conducted the Kolmogorov-Smirnov normality test on the Flickr and MS-COCO datasets. The results of this test, presented in Tab. 9, demonstrate that under a significance level of $\alpha = 0.05$, the scores of baselines conform to a normal distribution, thereby affirming the validity of our hypothesis.

Table 9: The normality tests on the membership scores of the baselines. The results show that, at a significance level of $\alpha = 0.05$, the baseline scores conform to a normal distribution.

| Method | MS-COCO | | | | Flickr | | | |
|---|---|---|---|---|---|---|---|---|
| | Member | | Hold-out | | Member | | Hold-out | |
| | Test Statistic | P-value | Test Statistic | P-value | Test Statistic | P-value | Test Statistic | P-value |
| Naive | 0.0285 | 0.8003 | 0.0435 | 0.2921 | 0.0279 | 0.8212 | 0.0464 | 0.2248 |
| SecMI | 0.0359 | 0.5295 | 0.0428 | 0.3104 | 0.0557 | 0.0862 | 0.0446 | 0.1031 |
| PIA | 0.0384 | 0.4406 | 0.0327 | 0.6467 | 0.0254 | 0.8948 | 0.0449 | 0.2587 |

**The Impact of Normality.** By comparing the p-values from the normality tests, we observe that normality does not significantly influence our method. As illustrated in Tab. 9, the baselines exhibit varying degrees of normality across different datasets, with SecMI occasionally demonstrating weaker normality and PIA's normality appearing somewhat inconsistent. Nevertheless, our approach is not affected by normality, as shown in Tab. 10.

Table 10: Impact of normality on our method. The results indicate that our method does not strongly rely on the normality assumption.

| Method | MS-COCO | | | Flickr | | |
|---|---|---|---|---|---|---|
| | ASR | AUC | TPR@1%FPR | ASR | AUC | TPR@1%FPR |
| SecMI | 82.09 | 89.37 | 16.79 | 71.49 | 77.31 | 6.19 |
| SecMI+F | 91.00 | 95.74 | 27.40 | 80.10 | 85.95 | 21.20 |
| Gain | +8.91 | +6.37 | +10.61 | +8.61 | +8.64 | +15.01 |
| PIA | 68.19 | 72.88 | 5.20 | 64.60 | 67.95 | 5.79 |
| PIA+F | 76.00 | 83.08 | 16.59 | 69.30 | 74.63 | 19.60 |
| Gain | +7.81 | +10.20 | +11.39 | +4.70 | +6.68 | +13.81 |

## D.3 DETAILED SETTINGS

As shown in Tab. 11, we present the segmentation of member and hold-out data across all datasets, ensuring that both member and hold-out data are independently and identically distributed, with equal quantities. Furthermore, we specify the batch size and the number of iterations for training across different datasets. Attacking under pre-trained conditions does not require additional training of the models.

Table 11: Detailed dataset settings and model settings.

| Model | Dataset | Resolution | Member | Hold-out | Batch-size | Iterations |
|---|---|---|---|---|---|---|
| DDIM | CIFAR-100 | 32 | 25000 | 25000 | 128 | 800000 |
| | STL10-U | 32 | 50000 | 50000 | 128 | 1600000 |
| | Tiny-ImageNet | 32 | 50000 | 50000 | 128 | 1600000 |
| Stable Diffusion v1.4 | Pokémon | 512 | 416 | 417 | 1 | 15000 |
| | Flickr | 512 | 1000 | 1000 | 1 | 60000 |
| | MS-COCO | 512 | 2500 | 2500 | 1 | 150000 |
| Stable Diffusion v1.4/5 | Laion-MI | 512 | 2500 | 2500 | / | / |

### D.4 ATTACK FOR PRE-TRAINED STABLE DIFFUSION

As shown in Tab. 12, when we tried to attack the pre-trained models, the effect of the filter was weak. This phenomenon can be attributed to the fact that the baselines completely failed in the pre-training setting. Their performance on ASR and AUC metrics was nearly equivalent to random guessing. The likely reason is that the assumptions of current attacks are not well-suited to the pre-training configuration. As a result, even when our filter was applied, it was difficult to achieve significant performance improvements.

Table 12: Under the pre-trained settings, the attack performance in stable diffusion.

| Method | SD1.4 | | | SD1.5 | | |
|---|---|---|---|---|---|---|
| | ASR | AUC | TPR@1%FPR | ASR | AUC | TPR@1%FPR |
| Naive | 53.19 | 52.82 | 0.20 | 53.86 | 52.69 | 0.20 |
| **Naive+F** | 53.50 | 53.09 | 0.20 | 54.10 | 53.07 | 0.20 |
| SecMI | 53.29 | 50.99 | 0.60 | 52.88 | 52.99 | 1.40 |
| **SecMI+F** | 54.50 | 51.85 | 1.60 | 54.30 | 53.43 | 1.80 |
| PIA | 52.99 | 52.04 | 0.20 | 53.67 | 53.22 | 0.40 |
| **PIA+F** | 53.10 | 52.11 | 0.20 | 53.70 | 53.24 | 0.40 |
| **Avg+** | **+0.54** | **+0.40** | **+0.33** | **+0.56** | **+0.28** | **+0.13** |

### D.5 THE GENERALITY OF HIGH-FREQUENCY FILTERING

The baseline methods (Naive, SecMI, and PIA) rely on the reconstruction errors of images when calculating membership scores, and our method significantly improves their performance. In fact, any attack that relies on the model's reconstruction capability can consider employing our method. We also add experimental results for CLID (Zhai et al., 2024), which is based on likelihood estimation. We included experiments on CLID under the typical strong overfitting setting in Tab. 13, where it achieves exceptionally high performance. In this setting, our method provides some improvement. Testing under the assumption of weak overfitting as outlined in Sec. 5.5. As shown in Tab. 14, it demonstrates that our method remains effective and shows significant performance improvements across different datasets. Overall, any methodology leveraging the model's capabilities for image processing can benefit from the frequency analysis and the techniques for mitigating high-frequency deficiency presented in this paper, as this flaw is inherently present in diffusion models.

Table 13: Attack performance of CLID under typical conditions.

| Method | Pokémon | | | MS-COCO | | | Flickr | | |
|---|---|---|---|---|---|---|---|---|---|
| | ASR | AUC | TPR@1%FPR | ASR | AUC | TPR@1%FPR | ASR | AUC | TPR@1%FPR |
| CLID | 97.75 | 99.39 | 77.00 | 98.50 | 99.43 | 90.00 | 93.25 | 96.97 | 84.00 |
| **CLID+F** | 98.25 | 99.57 | 77.00 | 98.62 | 99.51 | 92.00 | 95.50 | 98.38 | 90.00 |

Table 14: Attack performance of CLID under weak overfitting conditions.

| Method | Pokémon | | | MS-COCO | | | Flickr | | |
|---|---|---|---|---|---|---|---|---|---|
| | ASR | AUC | TPR@1%FPR | ASR | AUC | TPR@1%FPR | ASR | AUC | TPR@1%FPR |
| CLID | 87.71 | 91.14 | 54.20 | 89.52 | 93.32 | 59.75 | 88.76 | 92.00 | 56.00 |
| **CLID+F** | 89.75 | 93.73 | 60.25 | 92.10 | 95.41 | 66.00 | 91.20 | 94.53 | 64.00 |

### D.6 MEMBERSHIP SCORES DISTRIBUTION FOR SAMPLES FROM MEMBER AND HOLD-OUT SET.

As illustrated in Fig. 6, we further present the distribution of membership scores on the Pokémon and Flickr datasets. Additionally, we conducted a statistical analysis of the $\sigma_H/\sigma_M$. As shown in Tab.

15, the results strongly validate the effectiveness of our method: after applying the high-frequency filter, the $\sigma_H/\sigma_M$ exhibits varying degrees of improvement, which aligns closely with our theoretical expectations.

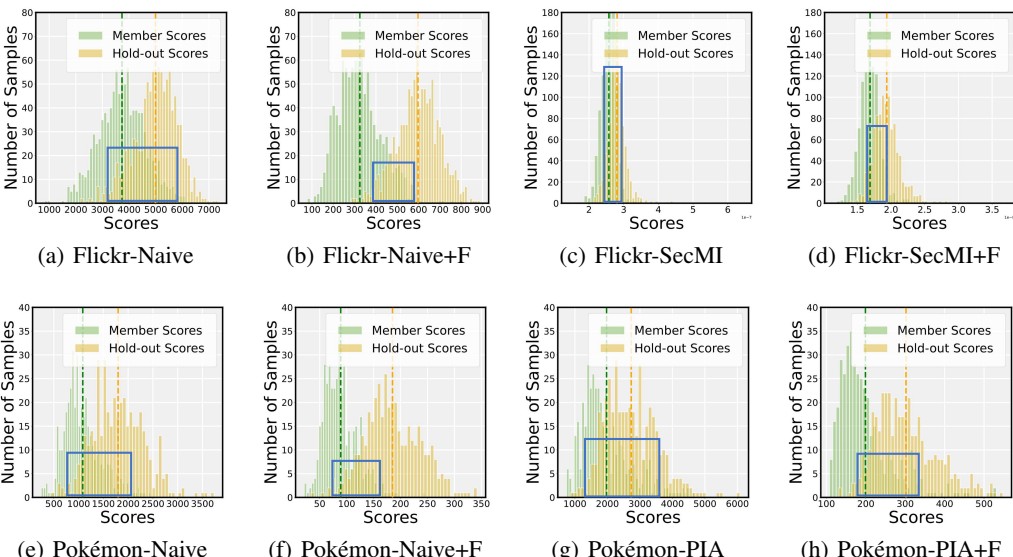

Figure 6: Membership scores distribution for samples from member set and hold-out set.

Table 15: Statistics of $\sigma_H/\sigma_M$ before and after applying filter to the baselines.

| Dataset | Metric | Naive | Naive+F | PIA | PIA+F | SecMI | SecMI+F |
|---------|--------|-------|---------|-----|-------|-------|---------|
| Pokémon | $\sigma_H/\sigma_M$ | 1.3541 | 1.5819 | 1.0453 | 1.1049 | 1.0841 | 1.3591 |
| MS-COCO | $\sigma_H/\sigma_M$ | 1.0896 | 1.7722 | 1.0121 | 1.2826 | 1.1969 | 1.6674 |

### D.7 MISSING TEXT

Previously, for attacks on stable diffusion, we typically assumed access to the text used for each image during training, which is a strong requirement for attackers. In real-world scenarios, we may not be able to obtain the text, which increases the difficulty of the attack. To simulate more realistic attack conditions, we conducted attacks in scenarios without text and with text generated by BLIP (Li et al., 2022). As shown in Tab. 16, when attacking without text or with the text generated by BLIP, although the performance of the baselines has declined noticeably, the filter still shows satisfactory results. This demonstrates that our method still exhibits good performance under more stringent attack conditions.

Table 16: Attack performance in no-text and BLIP-generated text conditions.

| Method | MS-COCO | | | | Flickr | | | |
|--------|---------|---|---|---|--------|---|---|---|
| | No Text | | BLIP Text | | No Text | | BLIP Text | |
| | ASR | AUC | ASR | AUC | ASR | AUC | ASR | AUC |
| Naive | 61.19 | 64.45 | 75.80 | 81.43 | 69.74 | 73.80 | 75.49 | 80.99 |
| **Naive+F** | 69.80 | 74.56 | 87.72 | 91.51 | 81.75 | 88.23 | 87.75 | 94.77 |
| SecMI | 69.60 | 74.77 | 81.85 | 87.99 | 75.00 | 79.22 | 77.99 | 84.00 |
| **SecMI+F** | 70.70 | 76.70 | 82.92 | 88.76 | 75.67 | 80.26 | 78.95 | 84.87 |
| PIA | 53.10 | 51.99 | 61.42 | 65.69 | 57.24 | 58.75 | 66.00 | 66.84 |
| **PIA+F** | 54.70 | 53.18 | 64.94 | 70.81 | 59.61 | 59.25 | 69.00 | 73.29 |
| **Avg+** | **+3.77** | **+4.41** | **+5.50** | **+5.32** | **+5.02** | **+5.32** | **+5.41** | **+7.03** |

## D.8 ABLATION STUDY ON DDIM

As shown in Tab. 17, ablation experiments were performed on DDIM using the Tiny-IN dataset, and the experimental phenomenon is similar to the results in fine-tuned stable diffusion. we recommend a value range for $r_t$ of $[3, 10]$ and for $s$ of $[0.0, 0.3]$. Within this range, the filter achieves optimal performance, significantly enhancing the baseline performance while exhibiting low sensitivity to changes in hyperparameters.

Table 17: Naive'attack performance at different $s$ and $r_t$ in Tiny-IN dataset.

| $s/r_t$ | $s = 0.0$ | | $s = 0.1$ | | $s = 0.2$ | | $s = 0.3$ | | $s = 0.4$ | | $s = 0.5$ | |
|---|---|---|---|---|---|---|---|---|---|---|---|---|
| | ASR | AUC | ASR | AUC | ASR | AUC | ASR | AUC | ASR | AUC | ASR | AUC |
| $r_t = 1$ | 70.64 | 77.41 | 80.16 | 87.79 | 78.90 | 86.41 | 78.19 | 85.62 | 77.89 | 85.27 | 77.77 | 85.09 |
| $r_t = 3$ | 83.14 | 90.32 | 83.65 | 90.95 | 83.98 | 91.38 | 83.12 | 90.68 | 81.86 | 89.40 | 80.57 | 88.13 |
| $r_t = 5$ | 85.05 | 92.20 | 85.13 | 92.28 | 85.03 | 92.22 | 84.43 | 91.87 | 83.57 | 91.11 | 82.56 | 90.06 |
| $r_t = 7$ | 84.62 | 91.87 | 84.59 | 91.83 | 84.39 | 91.66 | 83.87 | 91.33 | 83.36 | 90.78 | 82.59 | 90.02 |
| $r_t = 9$ | 82.68 | 90.14 | 82.62 | 90.09 | 82.51 | 89.94 | 82.21 | 89.67 | 81.84 | 89.29 | 81.31 | 88.78 |
| $r_t = 10$ | 81.74 | 89.21 | 81.66 | 89.16 | 81.54 | 89.02 | 81.29 | 88.81 | 80.93 | 88.49 | 80.56 | 88.07 |

In addition, we visualize ASR and AUC with different parameter settings. As illustrated in Fig. 7 our method demonstrates extreme robustness and is highly insensitive to hyperparameter variations.

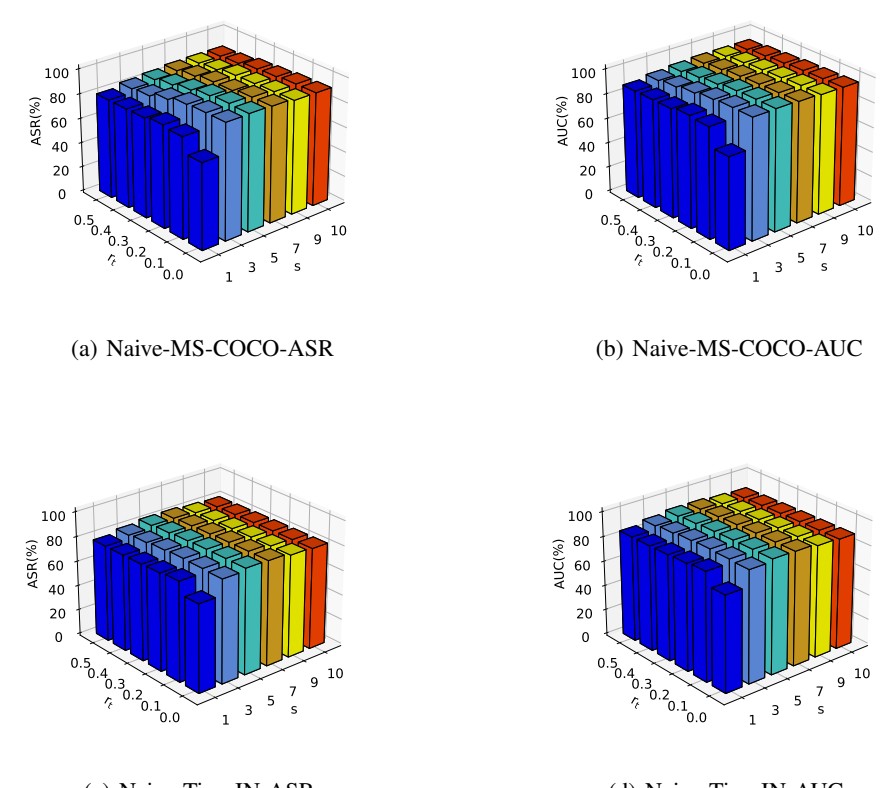

(a) Naive-MS-COCO-ASR

(b) Naive-MS-COCO-AUC

(c) Naive-Tiny-IN-ASR

(d) Naive-Tiny-IN-AUC

Figure 7: The three-dimensional histogram shows the ASR/AUC under different parameter settings. The x-axis represents the parameter $r_t$, the y-axis represents the parameter $s$, and the z-axis represents the ASR/AUC.

## D.9 ROC AND LOG-ROC CURVES VISUALIZATION

As illustrated in Fig. 8, We visualized the ROC and Log-ROC curves for different attacks on MS-COCO and Tiny-IN datasets. The blue curves and green curves represent the baselines before and after applying the high-frequency filter module. We can clearly see the powerful effect of the filter through the curves. At the same time, we visualize the ROC curves for different Naive parameter settings on MS-COCO and Tiny-IN datasets. The results are shown in Fig. 9, which show that our method provides a significant improvement over the baselines at different parameter settings.

## D.10 COMPUTE OVERHEAD

In this section, we assess the time cost of the filter. As shown in Tab. 18, we count the time spent on attacking all samples in the dataset, and the additional time overhead is approximately negligible by averaging on a single sample. Therefore, the experimental results show that our method hardly brings any additional time cost.

Table 18: We calculate the runtime of the attacks on the CIFAR-100 and Flickr datasets.

| Model | Dataset | Naive | Naive+F | PIA | PIA+F | SecMI | SecMI+F |
|---|---|---|---|---|---|---|---|
| DDIM | CIFAR-100 | 75s | 80s | 145s | 153s | 1470s | 1480s |
| Stable Diffusion | Flickr | 482s | 506s | 752s | 782s | 3474s | 3492s |

## D.11 THE IMPACT OF DEFENSE

We further present results on the Pokémon dataset to illustrate the impact of defense mechanisms on our method. As shown in Tab. 19 and Tab. 20, our method remains effective with the addition of defenses and provides significant improvements over the baselines.

Table 19: Attack performance ASR under the defenses. Our method remains effective and continues to deliver a clear performance improvement on ASR.

| $SS_{e_i}$ | DataAug | Naive | Naive+F | Gain | SecMI | SecMI+F | Gain | PIA | PIA+F | Gain |
|---|---|---|---|---|---|---|---|---|---|---|
| ✗ | ✗ | 80.00 | 92.00 | +12.00 | 81.08 | 86.00 | +4.92 | 75.49 | 87.00 | +11.51 |
| ✗ | ✓ | 79.50 | 87.88 | +8.38 | 76.37 | 83.75 | +7.38 | 72.27 | 80.87 | +8.60 |
| ✓ | ✗ | 67.91 | 73.12 | +5.21 | 77.52 | 84.01 | +6.49 | 63.32 | 68.61 | +5.29 |
| ✓ | ✓ | 65.75 | 70.13 | +4.38 | 74.12 | 79.00 | +4.88 | 59.12 | 64.63 | +5.51 |

Table 20: Attack performance TPR@1%FPR under the defenses. Our method remains effective and continues to deliver a clear performance improvement on TPR@1%FPR.

| $SS_{e_i}$ | DataAug | Naive | Naive+F | Gain | SecMI | SecMI+F | Gain | PIA | PIA+F | Gain |
|---|---|---|---|---|---|---|---|---|---|---|
| ✗ | ✗ | 6.99 | 46.00 | +42.01 | 15.30 | 33.00 | +17.70 | 10.60 | 38.00 | +27.40 |
| ✗ | ✓ | 6.49 | 41.25 | +34.76 | 12.74 | 31.25 | +18.51 | 7.75 | 39.25 | +31.50 |
| ✓ | ✗ | 6.38 | 26.41 | +20.03 | 12.87 | 25.12 | +12.25 | 6.37 | 18.74 | +12.37 |
| ✓ | ✓ | 5.24 | 20.50 | +15.26 | 11.24 | 22.75 | +11.51 | 5.24 | 13.75 | +8.51 |

## E VISUALIZATION OF HIGH-FREQUENCY FILTER EFFECTS.

We visualized the effects of high-frequency filtering in Fig. 10. Specifically, we presented the predicted images, along with their visualizations under the settings $(r_t = 3, s = 0.2)$ and $(r_t = 3, s = 0.0)$ in the Naive method.

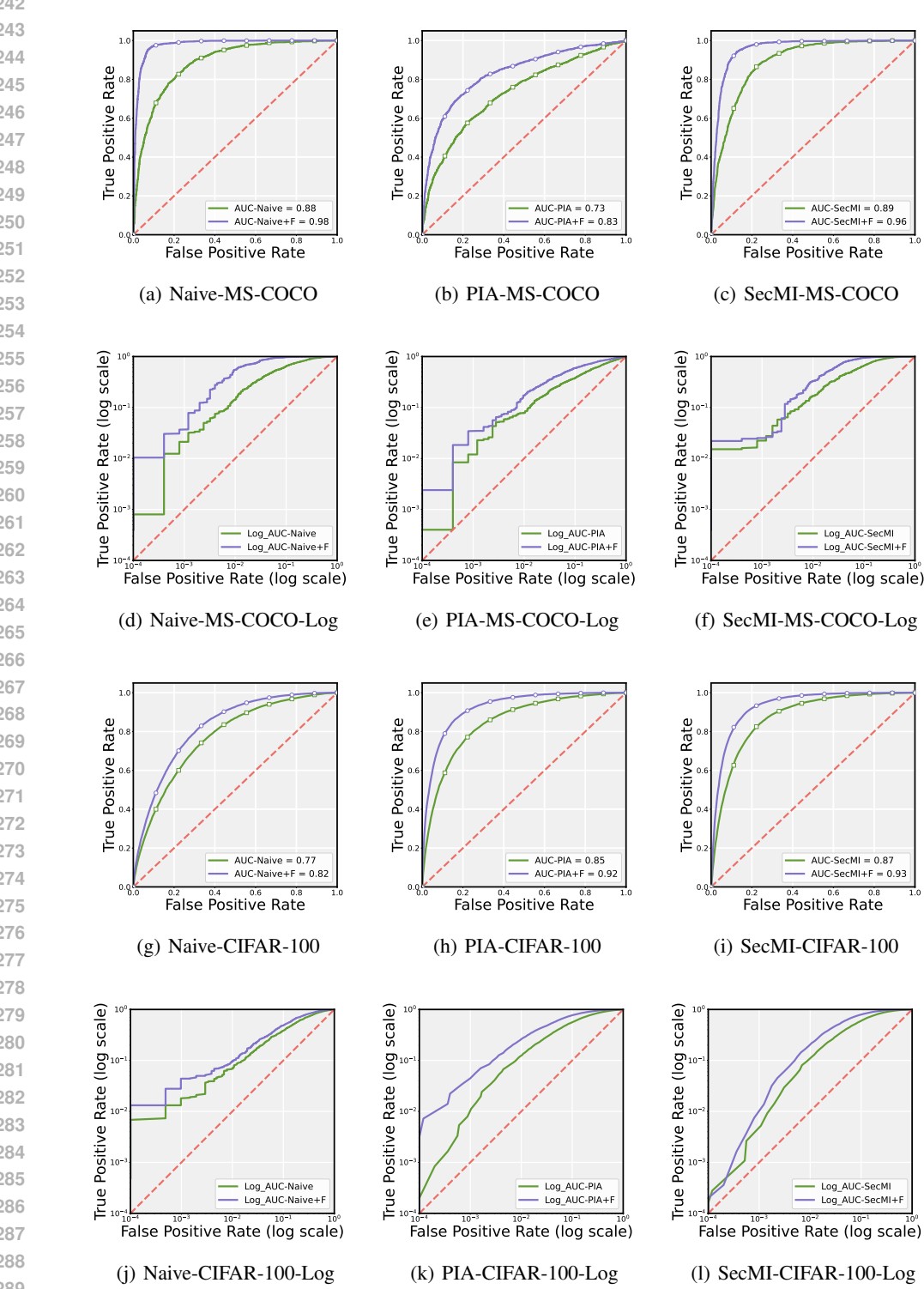

Figure 8: ROC and Log-ROC curves before and after applying the high-frequency filter for the baselines.

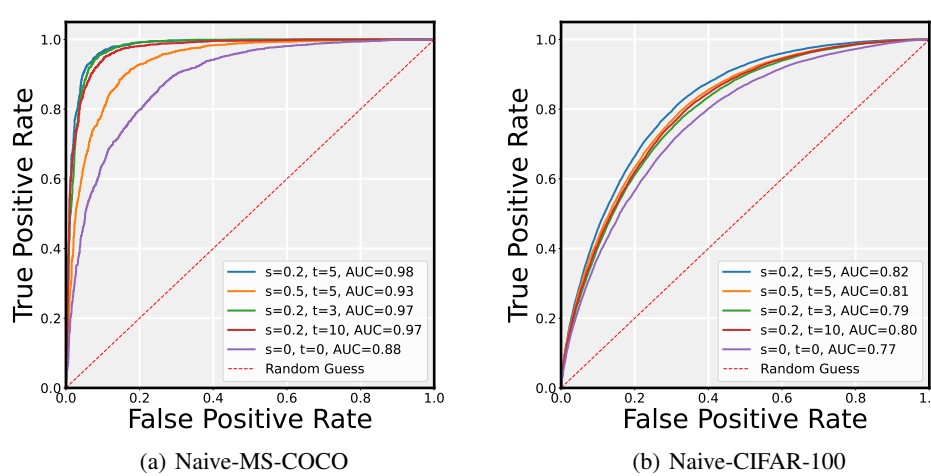

Figure 9: ROC curves of Naive with different parameter settings.

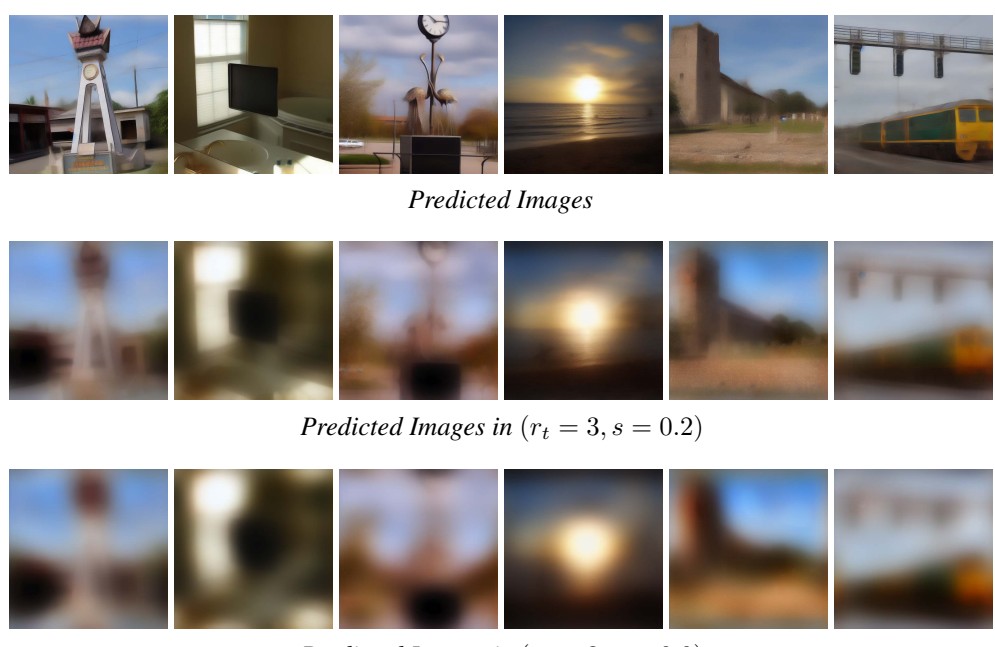

Figure 10: Visualization of High-Frequency Filter Effects in MS-COCO dataset.

## F ETHICS STATEMENTS

This study proposes a generalized membership inference attacks improvement algorithm aimed at enhancing the ability to infer whether specific samples were used during the training of diffusion models. Membership inference attacks have significant applications in unauthorized data usage audits and are one of the key means of maintaining copyright. Our method is expected to advance the development of copyright protection and model privacy research in the field of image generation. However, we also recognize that this method may pose privacy risks to existing diffusion models. To prevent the misuse of our research, all experiments in this study are conducted based on publicly available datasets and open-source model architectures. Additionally, the code for this research will be released to the public.

## G    THE USE OF LARGE LANGUAGE MODELS

During the preparation of this manuscript, we used a large language model (LLM) to improve the clarity and readability of the text. The use of the LLM was limited to language editing, and it did not contribute to the scientific content, analysis, or conclusions presented in this work. The authors take full responsibility for the originality and accuracy of the manuscript.

## H    REPRODUCIBILITY STATEMENT

We are committed to ensuring the reproducibility of our work. To this end, we provide the following information:

**Datasets:** All experiments in this study are conducted on publicly available datasets (e.g., MS-COCO, Pokemon, Flickr), ensuring that others can directly access the data we used.

**Models:** Our experiments are based on open-source diffusion model architectures, and no proprietary or restricted-access models are employed.

**Code and Implementation:** The implementation of our method, including preprocessing, training, and evaluation scripts, will be released to the public upon publication. We also provide the code in the supplementary materials.

**Hyperparameters and Settings:** Detailed hyperparameter configurations (e.g., batch size, learning rate, training iterations, $s$, $r_t$) are explicitly described in the paper and supplemental materials.

**Evaluation Metrics:** We report standard and widely adopted evaluation metrics (e.g., ASR, AUC, TPR@1%FPR) to ensure comparability with prior work.

