# OpenReview forum: "Unveiling Impact of Frequency Components on Membership Inference Attacks for Diffusion Models"
_ICLR.cc/2026/Conference — Submitted to ICLR 2026_

### Official Review · Reviewer_hx9c · 2025-10-29

**Soundness:** 1
**Presentation:** 2
**Contribution:** 1
**Rating:** 4
**Confidence:** 3

**Summary:**

This work is an analysis of the deficiencies of Diffusion Models when modeling high frequencies in the context of membership inference attacks. The authors claim that filtering out high-frequencies from suspected images can improve attack results by decrising the number of membership FN and thus improving TPR. They show detailed analysis and experimental results depicting improvement as well as ablation showing the impact of the masking strength on MIAs metrics.

**Strengths:**

- showing the correlation between membership scores and high-frequency content (shown for MS-COCO)
- describing how decreasing FN improves TPR and impacts Youden Index J (redefined as membership advantage in the paper - to my understanding); Overall linking improvements made (TPR -> J) with the effective differentiation of member and hold-out data (better classification).
- showing empirically benefits from masking high frequencies:  MIAs metrics show indirectly that membership FN decrease outweighs TN decrease when the high-frequency masking is used.
- showing that claimed benefits hold when two types of defenses against MIAs are adopted.

**Weaknesses:**

W1. RW does not mention any of works that analyze frequency components (not in DM not anywhere else) which is crucial to tell that mentioned frequency deficiencies insights are largely known and how new contribution fits to current knowledge. Authors only list previous works, they do not link previous work to their contribution.

W2. The improvement would be easier to understand if standard FN, FP, TPR, FPR metrics would be used. It is an additional work for the reader to understand how they are connected to terms used in the paper, while the idea is simple. To my understanding masking high-frequencies should decrease FP (good thing, the goal described in the paper), but at the same time should also decrease TN (bad thing, not commented nor shown). I understand that in tested settings FN decrease outweighs TN decrease but I would ask for clarification or TN results here.

W3. Tables descriptions are lacking details about the setup/ are not clear.
Table 1 - what are failed samples? Misclassified? Clarify the comparison what hold-out samples you compare with? Similar issues with Tab2, Tab5, Tab6.

W4. Results for 3 out of 4 grey-box methods mentions. Only partial results under one specific setup in the appendix for 4th method. Why?

W5. Other issues:
- the shown underlying problem with distribution shift is not novel for MIAs, yet no previous ideas for mitigations discussed in the paper.
- Contributions line 80 - std. dev of what? It is not specified by then.
- What is the definition of the term: error-based attacks? Are they attacks that use only reconstruction error? How general is the proposed General paradigm? It is created for error-based attacks and generalize only to error-based attacks?
- In my opinion LLMs were used extensively, especially in the first part of the paper, with rather mixed effect on the clarity, with uplifted words, LLM-sentences and genAI small problems popping here and there (e.g. lines 49-50, 162, 183, maybe also 323: Laion-MI mentioned with no results?).

**Questions:**

- weaknesses.
- 5.3 lines 380-381 - What is the generality of this claim, what are the assumptions?
- Fig2 - PIA - there is x scale difference for plots, improvement is not evident.
- What are the assumptions of Proposition 1? What is k?
- What does it mean in practice that s=0.1?
- Limitations: what is pre-training setting?

---

> ### Author Response · Authors · 2025-11-21
>
> Thanks for your acknowledgment of the **effective improvement in attack performance**, **robustness under defense** in our article. Below, we provide a detailed response to each comment:
>
>
> **W(1)**: The related work on the frequency domain components of diffusion models should be provided and introduced.
>
> **Answer(1)**: Thank you for your suggestion. We have added relevant literature and a brief introduction on frequency domain analysis of diffusion models in the related work section. For more details, please refer to the revised manuscript.
>
> **W(2)**: The improvement would be easier to understand if standard FN, FP, TPR, and FPR metrics would be used. To my understanding, masking high-frequencies should decrease FN (good thing), but at the same time should also decrease TN (bad thing).
>
> **Answer(2)**: Thank you for your suggestion. The metrics in this paper follow the settings of previous works [1][2][3][4]. We have provided additional statistics on TP, TN, FP, FN, TPR, and FPR on the Flickr dataset in Table_R(1). The results show a significant reduction in FP and FN, along with an increase in TP and TN. Therefore, our method does not cause a decrease in TN; rather, it increases it (good thing).
>
> **Table_R(1): TP, TN, FP, FN, TPR, and FPR on Flickr Dataset**
>
> | Method  | TP    | TN  | FP    | FN    | TPR     | FPR    |
> |---------|-------|-----|-------|-------|---------|--------|
> | Naive   | 805   | 766 | 234   | 195   | 0.805   | 0.234  |
> | Naive+F | 886   | 928 | 72    | 114   | 0.886   | 0.072  |
> | PIA     | 659   | 630 | 370   | 341   | 0.659   | 0.370  |
> | PIA+F   | 667   | 748 | 252   | 333   | 0.667   | 0.252  |
> | SecMI   | 816   | 703 | 297   | 184   | 0.816   | 0.297  |
> | SecMI+F | 837   | 818 | 182   | 163   | 0.837   | 0.182  |
>
> **W(3)**: The caption of the table should be more detailed and clear (Tab.1, Tab.2, Tab.5, Tab.6). Tab. 1 - what are failed samples? Misclassified? Clarify the comparison what hold-out samples you compare with?
>
> **Answer(3)**: The essence of membership inference attacks is to classify member and non-member data, so the failed samples are those that are misclassified. In Section 5.1, we have already explained that we randomly split the dataset into two equally sized parts: one consisting of members and the other of non-members (hold-out). We have added a detailed explanation in the caption of the table in the revised manuscript.
>
> **W(4)**: Results for 3 out of 4 grey-box methods mentions. Only partial results under one specific setup in the appendix for 4th method. Why?
>
> **Answer(4)**: CLID [4] is a method specifically designed for text-to-image diffusion. The baseline we presented is applicable in both conditional and unconditional settings. Moreover, CLID operates on a different principle from the baselines. It integrates multiple components and only indirectly leverages the image recovery capability. As a result, we conducted a separate analysis of CLID and demonstrated the effectiveness of our approach. We also included experiments on CLID under the typical strong overfitting setting in Table_R(2), where it achieves exceptionally high performance. In this setting, our method provides some improvement. Under weaker overfitting conditions, our method still results in clear improvements (Appendix D.5).
>
> **Table_R(2): CLID**
>
> | Dataset  | Method  | ASR    | AUC    | TPR@1%FPR  |
> |----------|---------|--------|--------|------------|
> | Pokemon  | CLID    | 97.75  | 99.39  | 77.00      |
> |          | CLID+F  | 98.25  | 99.57  | 77.00      |
> | MS-COCO  | CLID    | 98.50  | 99.43  | 90.00      |
> |          | CLID+F  | 98.62  | 99.51  | 92.00      |
> | Flickr   | CLID    | 93.25  | 96.97  | 84.00      |
> |          | CLID+F  | 95.50  | 98.38  | 90.00      |

---

> > ### Author Response · Authors · 2025-11-21
> >
> > **W(5.1)**: The problem with distribution shift is not novel for MIAs, yet no previous ideas for mitigations discussed in the paper.
> >
> > **Answer(5.1)**: In traditional MIAs research, distribution shifts [6][7][8][9] typically refer to mismatches between the global distributions of member and non-member data, such as significant differences between artificially constructed non-member datasets and the original training set, as well as label and attribute shifts [6]. In this paper, both member and non-member data are drawn from the same dataset, meaning there is no distribution shift. The phenomenon we observed can be explained by the membership scores of high-frequency components causing confusion and misclassification between member and non-member data. This indicates an inherent deficiency, which is largely unrelated to distribution shift. Additionally, previous research [6][7][8][9] on distribution shifts and mitigations has primarily focused on models like classification models and language models. Since diffusion models do not have components such as logits or confidence scores, prior studies on distribution shifts cannot be directly applied to diffusion models.
> >
> > **W(5.2)**: Contributions line 80 - std. dev of what? It is not specified by then.
> >
> > **Answer(5.2)**: This is the standard deviation of the membership score. We have made revisions to improve clarity in the revised manuscript.
> >
> > **W(5.3)**: What is the definition of the term: error-based attacks? Are they attacks that use only reconstruction error? How general is the proposed General paradigm? It is created for error-based attacks and generalize only to error-based attacks?
> >
> > **Answer(5.3)**: Yes, the error-based attacks use reconstruction error as the sole criterion for determining membership. Our paradigm is not limited to attacks that rely solely on reconstruction error; it can also be applied to attacks that indirectly or partially utilize reconstruction error. Our experiments on CLID further validate this perspective (CLID only indirectly utilizes the reconstruction error). With this paradigm, we are able to analyze attacks from a frequency domain perspective, and subsequently, examine the impact of high and low frequencies on previous attacks.
> >
> > **W(5.4)**: Some of the vocabulary used for LLM refinement should be modified. Laion-MI mentioned with no results?
> >
> > **Answer(5.4)**: In the Appendix of our paper, we have provided a statement regarding the use of LLM for language polishing. We modified certain wording in the paper to improve clarity, and the details can be found in the revised manuscript.  We mentioned in Section 5.2 that the results of Laion-MI are provided in Appendix D.4.

---

> > > ### Author Response · Authors · 2025-11-21
> > >
> > > **Q(1)**: Section 5.3 lines 380-381 - What is the generality of this claim, what are the assumptions?
> > >
> > > **Answer(Q1)**: This refers to our proposed Proposition 1, which demonstrates that filtering high-frequency components can amplify the advantage of the attack algorithm. The proposition applies to attacks that directly or indirectly utilize image reconstruction. The assumptions of Proposition 1 are that membership scores follow a normal distribution and the attack algorithm has a membership advantage $Adv^M(A)$. We clarified our descriptions in the revised manuscript to avoid misunderstandings.
> > >
> > > **Q(2)**: Fig2 - PIA - there is x scale difference for plots, improvement is not evident.
> > >
> > > **Answer(Q2)**: This section presents the visualization of membership scores, corresponding to the attack performance in Tab. 2. Both ASR and AUC show significant improvement. We have also included the results of the distribution overlap coefficient [4] in Table_R(3), where the overlapping portion is clearly reduced.
> > >
> > > **Table_R(3): Distribution Overlap Coefficient**
> > >
> > > | Dataset | Fig. 2(a): Naive | Fig. 2(b): Naive+F | Fig. 2(c): PIA | Fig. 2(d): PIA+F |
> > > |---------|------------------|--------------------|----------------|------------------|
> > > | Pokemon | 0.3942           | 0.2107             | 0.6406         | 0.4762           |
> > >
> > > **Q(3)**: What are the assumptions of Proposition 1? What is k?
> > >
> > > **Answer(Q3)**: The assumptions of Proposition 1 are that membership scores follow a normal distribution and the attack algorithm has a membership advantage $Adv^M(A)$. When $k$ exceeds a certain threshold, filtering the high-frequency components can amplify the advantages of the algorithm. $k = h_{M}/h_{H}$ represents the ratio of the standard deviation of membership scores in high-frequency components between member and non-member data. In the revised manuscript, we have clarified our descriptions to improve clarity and prevent potential misunderstandings.
> > >
> > > **Q(4)**: What does it mean in practice that s=0.1?
> > >
> > > **Answer(Q4)**: $s$ is the filtering factor, representing the intensity of the signal filtering. $s=0.1$ indicates that most of the signal is filtered, while $s = 0$ corresponds to complete filtering.
> > >
> > > **Q(5)**: What is pre-training setting?
> > >
> > > **Answer(Q5)**: We explained the pre-training setting in Section 5.1, which refers to performing attacks on the pre-trained model using the Laion-MI dataset. Laion-MI is a dataset specifically designed for membership inference attacks on pre-trained models.
> > >
> > > **References**
> > >
> > > [1] Jinhao Duan, Fei Kong, Shiqi Wang, et al. Are diffusion models vulnerable to membership inference attacks? In International Conference on Machine Learning, pages 8717–8730. PMLR, 2023.
> > >
> > > [2] Fei Kong, Jinhao Duan, RuiPeng Ma, et al. An efficient membership inference attack for the diffusion model by proximal initialization. arXiv preprint arXiv:2305.18355, 2023.
> > >
> > > [3] Tomoya Matsumoto, Takayuki Miura, and Naoto Yanai. Membership inference attacks against diffusion models. In 2023 IEEE Security and Privacy Workshops (SPW), pages 77–83. IEEE, 2023.
> > >
> > > [4] Shengfang Zhai, Huanran Chen, Yinpeng Dong, et al. Membership inference on text-to-image diffusion models via conditional likelihood discrepancy. Advances in Neural Information Processing Systems, 37:74122–74146, 2024.
> > >
> > > [5] Inman H F, Bradley Jr E L. The overlapping coefficient as a measure of agreement between probability distributions and point estimation of the overlap of two normal densities[J]. Communications in Statistics-theory and Methods, 1989, 18(10): 3851-3874.
> > >
> > > [6] Carlini N, Chien S, Nasr M, et al. Membership inference attacks from first principles[C]//2022 IEEE symposium on security and privacy (SP). IEEE, 2022: 1897-1914.
> > >
> > > [7] Yichuan S, Kotevska O, Reshniak V, et al. Assessing Membership Inference Attacks under Distribution Shifts[C]//2024 IEEE International Conference on Big Data (BigData). IEEE, 2024: 4127-4131.
> > >
> > > [8] Duan M, Suri A, Mireshghallah N, et al. Do membership inference attacks work on large language models?[J]. arXiv preprint arXiv:2402.07841, 2024.
> > >
> > > [9] Das D, Zhang J, Trantèr F. Blind baselines beat membership inference attacks for foundation models[C]//2025 IEEE Security and Privacy Workshops (SPW). IEEE, 2025: 118-125.

---

### Official Review · Reviewer_SxPG · 2025-10-31

**Soundness:** 3
**Presentation:** 3
**Contribution:** 3
**Rating:** 6
**Confidence:** 2

**Summary:**

The paper investigates the impact of frequency components on membership attacks on diffusion models. It provides a theoretical analysis to prove that filtering high-frequency components can boost membership attack performance. Experimental results on different datasets and models confirm the discovery.

**Strengths:**

- The paper investigates an overlooked component in membership attacks
- It provides a theoretical analysis to prove that filtering high-frequency components can boost attack performance
- The proposed approach consistently boost the attack performance across datasets and models

**Weaknesses:**

- In Proposition 1, why is "the standard deviation of membership scores" considered, instead of the mean scores?
- The high-frequency threshold $r_t$ is recommended to be in [3, 10].
  + Is it universal or resolution-dependent / dataset-dependent? An analysis of this is recommended to add.
  + This threshold is small, meaning most of the frequencies can be filtered out. For example, in Table 4, the configuration ($r_t = 3, s = 0$), e.g., removing all frequencies except the first three, could achieve ASR 93.70. This strong result is surprising and may indicate a weakness in the benchmark dataset.
- From Table 4, it seems $s = 0$ can provide competitive performance with other settings, while being simpler and probably more efficient. Why is the setting ($r_t = 5, s = 0.2$) selected as the default in Section 5?
- Section 5.6: The authors should investigate adaptive defenses that exploit knowledge of this high-frequency-based attack mechanism.

**Questions:**

- In Proposition 1, why is "the standard deviation of membership scores" considered, instead of the mean scores?
- The high-frequency threshold $r_t$ is recommended to be in [3, 10].
  + Is it universal or resolution-dependent / dataset-dependent? An analysis of this is recommended to add.
  + This threshold is small, meaning most of the frequencies can be filtered out. For example, in Table 4, the configuration ($r_t = 3, s = 0$), e.g., removing all frequencies except the first three, could achieve ASR 93.70. This strong result is surprising and may indicate a weakness in the benchmark dataset.
- From Table 4, it seems $s = 0$ can provide competitive performance with other settings, while being simpler and probably more efficient. Why is the setting ($r_t = 5, s = 0.2$) selected as the default in Section 5?
- Section 5.6: The authors should investigate adaptive defenses that exploit knowledge of this high-frequency-based attack mechanism.

---

> ### Author Response · Authors · 2025-11-21
>
> Thanks for your acknowledgment of the **provides a theoretical analysis**, **boost the attack performance across datasets and models** in our article. Below, we provide a detailed response to each comment:
>
> **W(1)**: In Proposition 1, why is "the standard deviation of membership scores" considered, instead of the mean scores?
>
> **Answer(1)**: A mean score alone reflects the model's average fit to member and non-member samples but fails to capture the degree of separation between the score distributions of the two classes. [1] defined the attack advantage $Adv^M(A)$ (i.e., the ability to distinguish between member and non-member). It is shown that the ratio of the standard deviations of the non-member and member distributions is positively correlated with the attack advantage. Inspired by [1], we use the standard deviation to measure the algorithm's advantage before and after filtering high-frequency components, i.e., Definition 1 and Eq. (8).
>
> **W(2.1)**: The high-frequency threshold $r_t$ is recommended to be in [3, 10]. Is it universal or resolution-dependent / dataset-dependent? An analysis of this is recommended to add.
>
> **Answer(2.1)**: This hyperparameter is universal. We conducted experiments on datasets with different resolutions, with results for lower-resolution datasets provided in Appendix D.8. Whether it’s 512×512 (high resolution) or 32×32 (low resolution), the frequency distribution of natural images follows a power-law pattern of "low frequencies dominating, high frequencies gradually decaying." The radius $r$ primarily reflects the level of pixel value changes, i.e., the frequency. The interval [3, 10] can be seen as a parameter setting that captures the common characteristics of the frequency distribution in natural images, making this setting universally applicable.
>
> **W(2.2)**: This threshold is small, meaning most of the frequencies can be filtered out. Tab.4, the configuration $(r_t=3, s=0)$, e.g., removing all frequencies except the first three, could achieve ASR 93.70. Does this suggest a weakness in the benchmark dataset?
>
> **Answer(2.2)**: It is important to clarify that when $r_t=3$, this threshold filters high-frequency signals while still preserving a substantial portion of the low-frequency information. To demonstrate this, we calculated the percentage of energy retention(%) in the frequency domain. As shown in Table_R(1), when $r_t=3$, the average energy retention rate reached $40%$, indicating that a significant portion of the frequency information is still retained. Therefore, this threshold is reasonable and does not suggest any issues with the dataset.
>
> **Table_R(1): Energy Retention When $r_t=3$**
>
> | Dataset  | Pokemon  | MS-COCO  | Flickr  |
> |----------|----------|----------|---------|
> | Energy   | 67.21%   | 40.43%   | 43.06%  |
>
> **W(3)**: From Table 4, it seems $s=0.0$ can provide competitive performance with other settings, while being simpler and probably more efficient. Why is the setting $(r_t=5, s=0.2)$ selected as the default in Section 5?
>
> **Answer(3)**: The computational cost is the same regardless of the specific hyperparameter settings.  We found that $(r = 5, s = 0.2)$ provides excellent performance across multiple datasets and attack methods. Additionally, $(r = 5, s = 0.0)$ is another highly effective hyperparameter combination, which further demonstrates the robustness of our method. We also emphasize in the paper that the range of hyperparameters is relatively broad.
>
> **W(4)**: The authors should investigate adaptive defenses that exploit knowledge of this high-frequency-based attack mechanism in Section 5.6.
>
> **Answer(4)**: We propose an adaptive defense strategy that applies random amplitude suppression, reducing the low-frequency components of images to 70-90% of their original values,  and adds noise with a variance of $0.05$ to the low-frequency domain during the model fine-tuning phase. The core mechanism of this strategy lies in moderately suppressing the representation weights of low-frequency components, thereby reducing the model's overfitting to low-frequency information in images. As shown in Table_R(2), experimental results demonstrate that this strategy mitigates the baselines and our method.
>
> **Table_R(2): With Adaptive Defense**
>
> | Method    | ASR        | AUC        | TPR@1%FPR  |
> |-----------|------------|------------|------------|
> | NA        | 74.34      | 81.12      | 5.06       |
> | NA+F      | 76.66      | 83.59      | 8.24       |
> | SecMI     | 73.13      | 78.59      | 3.37       |
> | SecMI+F   | 77.47      | 84.78      | 13.25      |
> | PIA       | 68.43      | 72.96      | 6.67       |
> | PIA+F     | 74.66      | 78.96      | 14.63      |
> | **Avg+**  | **+3.96**  | **+4.89**  | **+7.01**  |
>
>
> **References**
>
> [1] Yeom S, Giacomelli I, Fredrikson M, et al. Privacy risk in machine learning: Analyzing the connection to overfitting[C]//2018 IEEE 31st computer security foundations symposium (CSF). IEEE, 2018: 268-282.

---

> > ### Comment · Reviewer_SxPG · 2025-11-28
> >
> > Thank you the authors for the rebuttal. It has addressed most of my concerns.
> >
> > I am still not convinced by answers on Questions 2 and 3. Can you provide some qualitative examples showing the generated images before and after applying the proposed filtering mechanism, using both the $(r_t = 3, s = 0)$ and $(r_t = 3, s = 0.2)$, on a high-resolution dataset?

---

> ### Author Response · Authors · 2025-11-28
>
> Thank you for your suggestion. We provided the visualization results of MS-COCO dataset in Appendix E，where we presented the predicted images, along with their visualizations under the settings $(r_t=3, s=0.2)$ and $(r_t=3, s=0.0)$ in the Naive method. Details are provided in the revised manuscript.

---

### Official Review · Reviewer_rxDN · 2025-10-31

**Soundness:** 3
**Presentation:** 3
**Contribution:** 2
**Rating:** 6
**Confidence:** 4

**Summary:**

This paper investigates the impact of frequency components on membership inference attacks against diffusion models. The authors identify a "high-frequency deficiency" in how diffusion models process information, which causes current error-based MIAs to misclassify member data with high-frequency content as hold-out data and vice versa. To address this, they propose a high-frequency filter module that can be integrated into existing attacks to mitigate this deficiency. Extensive experiments on multiple datasets show that the filter improves attack performance.

**Strengths:**

1. The paper identifies the previously overlooked "high-frequency deficiency" in diffusion models, which adds a new dimension to understanding MIA limitations.
1. The proposed high-frequency filter method is simple, light-weighted, and effective to improve the performance of several current MIA methods.
2. The paper is easy to understand.

**Weaknesses:**

1. The study primarily relies on empirical observations of high-frequency signals' impact on MIA performance, lacking a rigorous theoretical exploration or analysis on why high-frequency components degrade attack efficacy, which can provide a more valuable insight for this phenomenon.
2. The ablation experiments in Section 5.4 show that suppressing all high-frequency signals harms performance, but the paper does not adequately explain why this occurs, and how to determine the optimal suppression threshold.
3. The paper only uses three tradition error-based MIA baselines. Can the method directly filter the high-frequency signals of the test images themselves instead of intermediate steps, so that this method can be applied to more types of advanced baselines? If not, it would be good to give some explanations.

**Questions:**

See the weaknesses.

---

> ### Author Response · Authors · 2025-11-21
>
> Thanks for your acknowledgment of the **simple, light-weighted, and effective method**, **adds a new dimension to understanding MIA limitations** in our article. Below, we provide a detailed response to each comment:
>
> **W(1)**: A theoretical analysis of why high-frequency components degrade attack performance is required.
>
> **Answer(1)**: Previous work [1] has also demonstrated, from the perspective of the power spectrum of natural images, that diffusion models initially learn to recover low-frequency components, and then, in fewer subsequent time steps, learn to recover high-frequency components. This results in greater uncertainty in the restoration of high-frequency details. Additionally, [2] empirically verifies, from a signal-to-noise ratio perspective, that during training, high-frequency components are more rapidly and early disrupted compared to low-frequency components. This leads to instability and greater variation in the quality of high-frequency generation.
>
> Our theoretical analysis is presented in **Proposition 1**. It shows that the variability and uncertainty of high-frequency components lead to an increased standard deviation in membership score, and filtering the standard deviation introduced by high-frequency components enhances the algorithm's advantage. It also indirectly demonstrates that high-frequency components diminishes the algorithmic advantage. This serves as the theoretical foundation for understanding the impact of high-frequency components on membership inference attack performance.
>
> **W(2)**: The ablation experiments in Section 5.4 show that suppressing all high-frequency signals harms performance, why this occurs? How to determine the optimal suppression threshold.
>
> **Answer(2)**: We recognize that this may lead to some misunderstanding. When the threshold is set too low, a significant amount of low-frequency information is also filtered out. Since most of the frequency information is removed, this naturally leads to a degradation in performance, which we also discuss in Section 5.4 of the paper.
>
> Regarding the selection of the optimal threshold, we determined it through experiments with various hyperparameters (including the threshold). In fact, within the range of hyperparameters (including the threshold) we recommend, excellent performance can be achieved.
>
> **W(3):** Can the proposed method be applied to more attacks? Can the method directly filter the high-frequency signals of the test images themselves instead of intermediate steps, so that this method can be applied to more types of advanced baselines?
>
> **Answer(3).1**: We further provide the advanced baseline CLID in Appendix D.5. Although CLID is not an error-based method, our approach remains effective because it indirectly leverages the model’s capability to reconstruct images. In fact,  any attack that relies on the image reconstruction capability of the model can potentially benefit from our method, as the high-frequency deficiency is inherent to diffusion models.
>
> **Answer(3).2**: Directly filtering the high-frequency signals of the test images themselves represents a black-box setting. Although our paper does not focus on black-box conditions, we have attempted to integrate our proposed method into previous black-box attacks [1]. As shown in Table R(1), the performance improvement after incorporating our method is relatively weak. Our analysis suggests that current black-box attacks primarily rely on semantic-level signals: they capture differences between member and non-member data through pre-trained feature extractors (such as DeiT, ViT, etc.), and determine membership based on the discriminability at the semantic representation level. Therefore, when applied to black-box attacks, as the frequency features optimized by our method have weak relevance to the semantic features relied upon by black-box methods, it does not offer significant gains for semantic-level judgments.
>
> **Table R(1): Black-Box Attack**
> | Method      | ASR    | AUC    | TPR@1%FPR  |
> |-------------|--------|--------|------------|
> | Black-T[3]  | 58.00  | 59.80  | 5.00       |
> | Black-T+F   | 58.13  | 60.11  | 5.00       |
> | Black-C[3]  | 62.50  | 66.40  | 11.00      |
> | Black-C+F   | 64.38  | 68.46  | 12.00      |
>
> **References**
>
> [1] Yang X, Zhou D, Feng J, et al. Diffusion probabilistic model made slim[C]//Proceedings of the IEEE/CVF Conference on computer vision and pattern recognition. 2023: 22552-22562.
>
> [2] Falck F, Pandeva T, Zahirnia K, et al. A Fourier Space Perspective on Diffusion Models[J]. arXiv preprint arXiv:2505.11278, 2025.
>
> [3] Yan Pang and Tianhao Wang. Black-box membership inference attacks against fine-tuned diffusion models. Network and Distributed System Security (NDSS) Symposium 2025.

---

### Official Review · Reviewer_ntyW · 2025-11-01

**Soundness:** 2
**Presentation:** 3
**Contribution:** 2
**Rating:** 2
**Confidence:** 4

**Summary:**

This paper investigates MIAs for diffusion models, mainly from the perspective of the frequency analysis. In detail, the authors observe that existing MIA methods suffer from a "high-frequency deficiency", where diffusion models handle high-frequency information inefficiently, leading to misclassification of member and nonmember. They formalize MIAs into a general paradigm based on reconstruction errors and propose a plug-and-play high-frequency filter module to mitigate this deficiency. Extensive experiments demonstrate that the filter enhances baseline attack performance across diverse datasets and models.

**Strengths:**

+ The paper is well-structured and clearly presented, making its contributions easy to follow.

+ The frequency perspective introduced a novel and valuable lens to analyze and understand the effectiveness of existing membership inference attacks (MIA).

**Weaknesses:**

+ (Major) The core contribution of analyzing memorization through frequency components (structure/low-frequency vs details/high-frequency) appears to have significant overlap with prior work. Specifically, [1] investigates membership inference by separating structural and detailed information, which is conceptually similar to the low-frequency/high-frequency components discussed here. The manuscript lacks a clear statement on its contribution beyond this established approach.

+ (Major) The underlying mechanism is still not clear. The explanation of **why** high-freq components are handled with more variation by diffusion models is insufficient. The high-level explanation in the introduction (Line 59-62) is vague and lacks support. Key questions remain unanswered: Is there any direct evidence or established literature demonstrating this behavior? What is the precise relationship between the observed "high-freq deficiency" and the well-known concept of "spectral bias" in neural network [2] ?

+ (Major) The derivation and presentation of Proposition 1 are confusing. The condition $k^2>1+\frac{2 \Delta}{h^2_H}(...)$ seems to be presented as a premise, but the proof in Appendix C suggests it is actually a conclusion. The logic appears to be: if $\frac{\sigma_H'}{\sigma_M'}>\frac{\sigma_H}{\sigma_M}$, then we have $k^2>1+\frac{2 \Delta}{h^2_H}(...)$. Besides, why the assumption $k^2>1+\frac{2 \Delta}{h^2_H}(...)$ holds? Any empirical evidence?

+ (Major) To thoroughly demonstrate the method's effectiveness, especially in the critical low-confidence scenario, it is essential to provide ROC curves on a ​​log-scaled axis​​. Linear-scale ROC curves can make performance look artificially strong by compressing the low false-positive rate region, and a log scale is the standard for a more rigorous evaluation.

+ (Minor) The proof (Appendix B) showing that "error" equals "sample" is considered trivial. Furthermore, a very similar proof has already been provided by [3] (equations 7 and 14).

+ (Minor) The method to obtain the threshold used in the threat model is not described. Please specify how this threshold is determined.

[1] Unveiling structural memorization: Structural membership inference attack for text-to-image diffusion models. ACM MM 2024

[2] On the Spectral Bias of Neural Networks. ICML 2019

[3] Are Diffusion Models Vulnerable to Membership Inference Attacks. ICML 2023

**Questions:**

See the weaknesses.

---

> ### Author Response · Authors · 2025-11-21
>
> Thanks for your acknowledgment of the **well-structured and clearly presented writing**, **novel and valuable perspective** in our article. Below, we provide a detailed response to each comment:
>
> **W(1)**: A clear statement of contributions beyond [1] is needed.
>
> **Answer(1)**: Thanks for your suggestions. The differences of contribution between our work and [1] can be summarized in three **key points**:
>
> - Fundamental differences in the analysis focus: [1] focuses on the human-made separation of "semantic structure," while we focus on the physical characteristics of "frequency-domain components." Simultaneously, structural information is generally closer to low frequencies, but it includes critical high-frequency information (edges); detail information is typically high-frequency, but it does not necessarily equate to all high-frequency components. Therefore, the two (structure/details vs. low-/high-frequency ) are not in correspondence.
> - Differences in the generalizability of findings: The findings in [1] are specific to DDIM inversion, whereas our findings, the high-frequency deficiency, can be broadly applicable to various membership inference attacks on diffusion models [5][6][7][8].
> - Differences in the generalizability of methods: [1] designs a targeted attack, while our method provides a unified enhancement to various existing attacks, offering broader adaptability.
>
> Our analysis reveals that "the interference of high-frequency components on attack performance is a common mechanism across attacks." Based on this analysis, we propose a general improvement that can enhance the performance of existing error-based attacks. The core contribution of [1] focuses on the structural similarity during the DDIM inversion process. Specifically, it identifies that member data retains a more complete structure during DDIM inversion, based on this observation, proposes a targeted attack method based on "structural similarity."
>
> **W(2)**: Can the underlying mechanism of the handling of high-frequency components by diffusion models be explained more comprehensively? Is there existing literature or direct evidence supporting this? What is the relationship between the "high-freq deficiency" and the concept of "spectral bias" in neural networks [2]?
>
> **Answer(2).1**: Previous work [3] has also demonstrated, from the perspective of the power spectrum of natural images, that diffusion models initially learn to recover low-frequency components, and then, in fewer subsequent time steps, learn to recover high-frequency components. This results in greater uncertainty in the restoration of high-frequency details. Additionally, [4] empirically verifies, from a signal-to-noise ratio perspective, that during training, high-frequency components are more rapidly and early disrupted compared to low-frequency components. This leads to instability and greater variation in the quality of high-frequency generation.
>
> **Answer(2).2**: In deep neural networks, the frequency principle is a commonly observed phenomenon: models tend to first fit low-frequency components, followed by a gradual adaptation to higher-frequency components [2]. The diffusion model leverages neural networks to predict noise, and as such, is also influenced by the "spectral bias" of the neural network. Therefore, "spectral bias" could be one of the reasons behind the "high-frequency deficiency" of diffusion models. We suggest that the high-frequency deficiency is caused, on one hand, by the training process of the diffusion model, and on the other hand, potentially by the "spectral bias" of the neural networks.
>
> We provide an additional introduction on the frequency-domain processing of diffusion models in **Sec. 2 (RELATED WORK)** in the revised manuscript.

---

> ### Author Response · Authors · 2025-11-21
>
> **W(3)**: The derivation and presentation of Proposition 1 should be further improved. Why the assumption $k^2 > 1+\frac{2\Delta}{h_H^2}(l_{M} + 2\Delta -\sqrt{(l_{M}+2\Delta)^2 + h_H^2})$ holds? Any empirical evidence?
>
> **Answer(3).1**: Thank you for your suggestion. The proof of Proposition 1 is essentially an equivalent derivation rather than a deduction that assumes the conclusion. To avoid any potential misunderstanding, we have revised the proof of Proposition 1 in the revised manuscript.
>
> **Answer(3).2**: $k = h_{M}/h_{H}$ represents the ratio of the standard deviation of membership scores in high-frequency components between member and non-member data. The assumption indicates that when this ratio exceeds a specific threshold, filtering high-frequency components will amplify the algorithm’s advantage. We have provided experiments in Appendix D.2, demonstrating that $k^2 > 1+\frac{2\Delta}{h_H^2}(l_{M} + 2\Delta -\sqrt{(l_{M}+2\Delta)^2 + h_H^2)}$ holds across all datasets. The value of $k$ typically hovers around 1. Consequently, $k^2$ is generally close to 1. In contrast, the threshold term on the right-hand side of the inequality, i.e., $1+\frac{2\Delta}{h_H^2}(l_{M} + 2\Delta -\sqrt{(l_{M}+2\Delta)^2 + h_H^2})$ is inherently less than 1. Therefore, our assumption generally holds.
>
> **W(4)**: It is essential to provide ROC curves on a log-scaled axis.
>
> **Answer(4)**: We have provided ROC curves on a log-scaled axis in Appendix D.9. The details are provided in the revised manuscript. Our method achieves a good performance improvement at an extremely low FPR.
>
> **W(5)**: What is the difference between the proof in Appendix B, which shows that 'error' equals 'sample,' and the proof already provided in [5] ?
>
> **Answer(5)**:  Our proof differs from the one provided in [5], as the purposes and processes of the two proofs are distinct. Our proof focuses on the integration and transformation of error-based methods [5][6][7], whereas [5] aims to validate SecMI [5] and perform error analysis. The main contribution of our proof in Appendix B is to unify previous methods [5][6][7] based on reconstruction error by transforming them into a distance between the denoised image and the original image. This transformation allows us to analyze image recovery from a frequency-domain perspective, which in turn enables the examination of the impact of high- and low-frequency components on the attacks. The proof in [5] shows that the SecMI method approximates the distance between the ground-truth image at step $t$ and the model's prediction of it at step $t$.
>
> **W(6)**: Please specify how this threshold is determined.
>
> **Answer(6)**: Since our method is a unified improvement of previous attacks, we follow the threshold acquisition procedure described in the publications of these baselines. In brief, the optimal threshold is determined by training a shadow model on identically distributed data (which is explicitly partitioned into member and non-member sets) and is then applied to the target model.
>
> **References**
>
> [1] Qiao Li, Xiaomeng Fu, Xi Wang, et al. Unveiling structural memorization: Structural membership inference attack for text-to-image diffusion models. In Proceedings of the 32nd ACM International Conference on Multimedia, pages 10554–10562, 2024.
>
> [2] Rahaman N, Baratin A, Arpit D, et al. On the spectral bias of neural networks[C]//International conference on machine learning. PMLR, 2019: 5301-5310.
>
> [3] Yang X, Zhou D, Feng J, et al. Diffusion probabilistic model made slim[C]//Proceedings of the IEEE/CVF Conference on computer vision and pattern recognition. 2023: 22552-22562.
>
> [4] Falck F, Pandeva T, Zahirnia K, et al. A Fourier Space Perspective on Diffusion Models[J]. arXiv preprint arXiv:2505.11278, 2025.
>
> [5] Jinhao Duan, Fei Kong, Shiqi Wang, et al. Are diffusion models vulnerable to membership inference attacks? In International Conference on Machine Learning, pages 8717–8730. PMLR, 2023.
>
> [6] Fei Kong, Jinhao Duan, RuiPeng Ma, et al. An efficient membership inference attack for the diffusion model by proximal initialization. arXiv preprint
> arXiv:2305.18355, 2023.
>
> [7] Tomoya Matsumoto, Takayuki Miura, and Naoto Yanai. Membership inference attacks against diffusion models. In 2023 IEEE Security and Privacy Workshops (SPW), pages 77–83. IEEE, 2023.
>
> [8] Shengfang Zhai, Huanran Chen, Yinpeng Dong, et al. Membership inference on text-to-image diffusion models via conditional likelihood discrepancy.
> Advances in Neural Information Processing Systems, 37:74122–74146, 2024.

---

### Author Response · Authors · 2025-11-26

Thank you to all the reviewers for their positive feedback and constructive suggestions. We have updated our manuscript with the following modifications, which are highlighted in **blue**：

- **Section 2:** We have included a discussion on related work regarding Frequency Analysis for Diffusion Models.
- **Section 4.2:** We have added a description of the failed samples in Tab. 1.
- **Section 5.6:** We have added experiments for the adaptive defense section.
- **Appendix C:** We have revised the proof of Proposition 1.
- **Appendix D.5:**  We have added experiments for CLID under typical settings.
- **Appendix D.9:**  We have added ROC curves on a log-scaled axis.
- **Appendix E:** We have presented the predicted images, along with their visualizations under the settings $(r_t=3, s=0.2)$ and $(r_t=3, s=0.0)$.
- **Other:** We have revised certain descriptions in the manuscript to improve clarity.

---

### Author Response · Authors · 2025-12-03
**Summary (1/3)**

We thank all reviewers for their constructive feedback. We are encouraged by the positive remarks on our **novel and valuable perspective** (Reviewers ntyW, rxDN, SxPG), the **strong and robust performance** (Reviewers rxDN, SxPG, hx9c), and the **clear writing and well-organized structure** (Reviewers ntyW, rxDN).

We summarize the **main contributions** of this work as follows:

- **Insight:** This study is the first to explore the impact of frequency domain information on MIAs targeting diffusion models. We formalize a general paradigm for existing error-based attacks and conduct an in-depth analysis of the impact of frequency domain information. The results reveal that existing attacks generally overlook the effects of "high-frequency deficiency", which limits their performance.
- **Method:** To address this limitation, we propose a plug-and-play high-frequency filter module. This module effectively suppresses "high-frequency deficiency", and we theoretically demonstrate its capacity to improve attack advantage. This module can be seamlessly integrated into the error-based attacks with negligible additional time overhead.



### **Below, we provide a summary of the rebuttal:**

**About Reviewer rxDN (Rating: 6; Confidence: 4):**

We have provided detailed, point-by-point responses to all of reviewer **rxDN**’s comments and questions, and have incorporated the relevant experimental results. Due to system constraints, we have not yet received further feedback from reviewer **rxDN**. Below is a summary of our responses:

**(1)**: A theoretical analysis of why high-frequency components degrade attack performance is required.

- We explain the characteristics and underlying reasons for how diffusion models handle high-frequency components, and we further explain that Proposition 1 validates that filtering high-frequency amplifies the advantage of attack algorithms. This provides a theoretical analysis of how high-frequency components influence attack performance.

**(2)**: The ablation experiments show that suppressing all high-frequency signals harms performance. Why does this occur? How to determine the optimal suppression threshold.

- We clarify the misunderstanding regarding the ablation study. When $r_t$ is set to a very small value ($r_t=1.0$), most frequency components are removed, so the performance reduction is expected. Regarding the optimal threshold, we determined it through experiments with various hyperparameters (including the threshold).

**(3):** Can the proposed method be applied to more attacks? Directly filter the high-frequency signals of the test images themselves instead of intermediate steps (Black-Box)?

- We clarify the broad applicability of our method, noting that any attack that relies on image reconstruction can benefit from it. In addition, we have included results for attacks evaluated in a black-box setting.



**About Reviewer SxPG (Rating: 6; Confidence: 2):**

We have provided detailed, point-by-point responses to all of Reviewer **SxPG**’s comments and questions, and have incorporated the relevant experimental results. We have addressed the reviewer’s questions and added high-frequency filtering visualizations in response to their feedback. Due to system constraints, we have not yet received further feedback from reviewer **SxPG**. Below is a summary of our responses:

**(1)**: In Proposition 1, why is "the standard deviation of membership scores" considered?

- We justify the use of standard deviation as a measure of attack advantage based on its statistical properties and insights from prior work.

**(2)**: The range of $r_t$ is universal or resolution-dependent / dataset-dependent? The configuration $(r_t=3, s=0)$, e.g., removing all frequencies except the first three, could achieve ASR 93.70. Does this suggest a weakness in the benchmark dataset?

- The hyperparameter is universal. We clarify the broad applicability of our hyperparameter range and explain that even with $r_t=3.0, s=0.0$, the image still retains a substantial amount of low-frequency information. We quantify the retained information before and after filtering, and we have also added visualizations corresponding to the setting $r_t=3.0, s=0.0$ in the revised manuscript.

**(3)**: Why is the setting $(r_t=5, s=0.2)$ selected as the default?

- We explain our choice of hyperparameters and emphasize that our method is robust to these settings, achieving strong performance across a wide range of values.

**(4)**: Need to investigate adaptive defenses.

- We have incorporated additional experiments on adaptive defenses as suggested by the reviewer, and the corresponding results are presented in the revised manuscript.

---

> ### Author Response · Authors · 2025-12-03
> **Summary (2/3)**
>
> **About Reviewer hx9c (Rating: 4; Confidence: 3):**
>
> We have provided detailed, point-by-point responses to all of reviewer **hx9c**’s comments and questions, and have incorporated the relevant experimental results. Due to system constraints, we have not yet received further feedback from reviewer **hx9c**. Below is a summary of our responses:
>
> **(1)**: The related work on the frequency domain components of diffusion models should be provided and introduced.
>
> - We have incorporated the relevant discussion and supporting explanations into the revised manuscript.
>
> **(2)**: FN, FP, TPR, and FPR metrics should be used.
>
> - We have included experiments and the results demonstrate that our method substantially increases both FP and FN, which contradicts the reviewer’s expectation.
>
> **(3)**: What are failed samples? Clarify the comparison what hold-out samples you compare with?
>
> - We provide a detailed explanation of what a failed sample is, and have added clarification in Tab. 1 of the manuscript. In addition, we explain the partition between the member and hold-out samples.
>
> **(4)**: Results for 3 out of 4 grey-box methods mentioned. Only partial results under one specific setup are in the appendix for the CLID method. Why?
>
> - We explain that the principle of CLID [1] differs from the baselines discussed in the main text, as it only indirectly leverages the model’s image reconstruction capability. We have also included experiments on CLID under the standard setting.
>
> **(5)**: No prior ideas for mitigating distribution shift are discussed.
>
> - We clarify that the distribution shift is largely unrelated to our work, as neither prior studies nor our experimental settings show any evidence of distribution shift.
>
> **(6)**: Fig2 - improvement is not evident.
>
> - We highlight the improvement in attack performance shown in Fig. 2 and quantify this enhancement using the overlap coefficient.
>
> **(7)**: What is the definition of the term: error-based attacks? How general is the proposed general paradigm?
>
> - We explain the definition and generality of error-based attacks, and emphasize that our general paradigm and method can be applied to any attack that directly or indirectly leverages the model’s image reconstruction capability.
>
> **(8):** Clarification and definition of terms.
>
> - We have provided the reviewer with explanations regarding the standard deviation, Laion-MI results, the assumption in Proposition 1, the definition of $k$, the significance of $s=0.1$, and the pre-training setting. Additionally, we have revised certain wording in the manuscript to improve clarity.
>
>
>
> **[1]** Membership inference on text-to-image diffusion models via conditional likelihood discrepancy.

---

> ### Author Response · Authors · 2025-12-03
> **Summary (3/3)**
>
> **About Reviewer ntyW (Rating: 2; Confidence: 4):**
>
> We have provided detailed, point-by-point responses to all of reviewer **ntyW**’s comments and questions, and have incorporated the relevant experimental results. Due to system constraints, we have not yet received further feedback from reviewer **ntyW**. Below is a summary of our responses:
>
> **(1)**: A clear statement of contributions beyond [2] is needed.
>
> - We respectfully note that the concerns regarding our contribution were raised from a misunderstanding of the conceptual distinctions between our work and [2]. As clarified in our response, [2] studies structural similarity in DDIM inversion based on the semantic structure of image regions, while our work fundamentally analyzes frequency-domain components and reveals a broader high-frequency deficiency that affects various existing membership inference attacks. The differences of contribution between our work and [2] can be summarized in three **key points**: **(1)** fundamental differences in the analysis focus, **(2)** differences in the generalizability of findings, **(3)** differences in the generalizability of methods.
>
> **(2)**: The underlying mechanism of how diffusion models handle high-frequency components should be explained more comprehensively.
>
> - We explain the underlying reasons why diffusion models exhibit varying behaviors in handling high-frequency components, and we have incorporated the relevant discussion and supporting explanations into the revised manuscript.
>
> **(3)**: The derivation and presentation of Proposition 1 should be further improved. Why the assumption $k$ holds? Any empirical evidence?
>
> - We clarify that the proof of Proposition 1 is essentially an equivalent derivation rather than a deduction that assumes the conclusion. To avoid any potential misunderstanding, we have revised the proof of Proposition 1 in the revised manuscript. The empirical analysis supporting the assumption regarding $k$ is provided in the Appendix.
>
> **(4)**: Need ROC curves on a log-scaled axis.
>
> - We have provided ROC curves on a log-scaled axis in the Appendix. The details are provided in the revised manuscript.
>
> **(5)**: What is the difference between the proof in Appendix B and the proof provided in [3]?
>
> - We emphasize that the derivation in Appendix B represents a transformation of prior methods, specifically converting from a noise perspective to an image perspective, and is therefore conceptually distinct from previous proofs.
>
> **(6)**: How is the threshold determined?
>
> - We explain the procedure for determining the threshold, which strictly follows the settings in the original baseline papers.
>
>
>
> **[2]** Unveiling structural memorization: Structural membership inference attack for text-to-image diffusion models.
>
> **[3]** Are diffusion models vulnerable to membership inference attacks?

---

### Meta-Review · Area_Chair_nMUE · 2026-01-07

**Summary:**

The paper proposed a detailed study on the frequency's influence on MIA performance. They find that existing MIA methods suffer from a high frequency deficiency, where diffusion models handle high-frequency information inefficiently, leading to misclassification of member and nonmember. They prove their finding theoretically and empirically. To tackle this problem, they propose a plug-and-play high-frequency filter module.

Strengths:

1. The paper is well-structured, its proof and experiments are reasonable.

2. The perspective is interesting.

Weaknesses:

1. If the paper is a purely analytical paper, a more detailed analysis of its reason and empirical results is needed to prove that the high-frequency deficiency is inevitable due to certain training features or datasets.

2. If we treat the paper as a new methodology, its weakness is no guarantee or guideline for certain hyper-parameter selection. Tuning hyperparameters with other datasets is not reasonable in the auditing setting as different datasets and training settings may have different features. And it will weaken the credibility of the auditing, as changing a different hyper-parameter may make the prediction change.

Therefore, I think the paper still needs to be modified before its acceptance. I recommend rejection on this paper.

**Reviewer Concerns:**

The reviewers are mostly concerned with its novelty, theoretical analysis, hyper-parameter selection, empirical settings and etc. I think the most important concerns are: reasons for the finding and hyper-parameter selection are not well-addressed.

**Reviewer Scores:**

I think the negative reviewer will not change their mind as the key concerns still remains.

---

### Decision · Program_Chairs · 2026-01-26

Reject